# Generation of mature compact ventricular cardiomyocytes from human pluripotent stem cells

Shunsuke Funakoshi[1], Ian Fernandes[1,2], Olya Mastikhina[3], Dan Wilkinson[4], Thinh Tran[5], Wahiba Dhahri[1], Amine Mazine[1,6], Donghe Yang[1,2], Benjamin Burnett[4], Jeehoon Lee[4], Stephanie Protze[1,5], Gary D. Bader [5,7,8], Sara S. Nunes [3,6,9,10], Michael Laflamme[1,9,11] & Gordon Keller [1,2✉]

Compact cardiomyocytes that make up the ventricular wall of the adult heart represent an important therapeutic target population for modeling and treating cardiovascular diseases. Here, we established a differentiation strategy that promotes the specification, proliferation and maturation of compact ventricular cardiomyocytes from human pluripotent stem cells (hPSCs). The cardiomyocytes generated under these conditions display the ability to use fatty acids as an energy source, a high mitochondrial mass, well-defined sarcomere structures and enhanced contraction force. These ventricular cells undergo metabolic changes indicative of those associated with heart failure when challenged in vitro with pathological stimuli and were found to generate grafts consisting of more mature cells than those derived from immature cardiomyocytes following transplantation into infarcted rat hearts. hPSC-derived atrial cardiomyocytes also responded to the maturation cues identified in this study, indicating that the approach is broadly applicable to different subtypes of the heart. Collectively, these findings highlight the power of recapitulating key aspects of embryonic and postnatal development for generating therapeutically relevant cell types from hPSCs.

[1] McEwen Stem Cell Institute, University Health Network, Toronto, ON, Canada. [2] Department of Medical Biophysics, University of Toronto, Toronto, ON, Canada. [3] Toronto General Hospital Research Institute, University Health Network, Toronto, ON, Canada. [4] BlueRock Therapeutics, New York, NY, USA. [5] Department of Molecular Genetics, University of Toronto, Toronto, ON, Canada. [6] Institute of Biomedical Engineering, University of Toronto, Toronto, ON, Canada. [7] Department of Computer Science, University of Toronto, Toronto, ON, Canada. [8] The Donnelly Centre, University of Toronto, Toronto, ON, Canada. [9] Laboratory of Medicine and Pathobiology, University of Toronto, Toronto, ON, Canada. [10] Heart & Stroke/Richard Lewar Centre of Excellence, University of Toronto, Toronto, ON, Canada. [11] Peter Munk Cardiac Centre, University Health Network, Toronto, ON, Canada. ✉email: gordon.keller@uhnresearch.ca

Human pluripotent stem cell (hPSC)-derived cardiomyocytes represent a promising source of cells for developing in vitro models to study diseases of the human heart and for establishing therapies to treat them. Successful translation of these applications to clinical and/or commercial practice is, however, dependent on our ability to derive the appropriate target cell(s) of the disease of interest from hPSCs in vitro[1]. While diseases can affect different cell types of the heart, those that impair ventricular function such as myocardial infarction (MI) and ventricular arrhythmias are amongst the most severe and debilitating and can lead to organ dysfunction and failure. Given this, hPSC-derived ventricular cardiomyocytes are attractive therapeutic targets for both modeling life threatening arrhythmias and heart failure in vitro and for developing cell therapies to remuscularize ventricular myocardium damaged by MI. However, to produce appropriate ventricular cells for these applications, it is essential to understand how they are specified and develop in the embryo and neonate and apply this knowledge to the generation of comparable cell types from hPSCs.

In the mouse, the cardiomyocytes that make up the ventricular chambers derive from progenitors that are specified at the earliest stages of heart development, prior to the emergence of progenitors that contribute to the atrial and sinoatrial pacemaker lineages[2]. In addition to distinct right and left ventricular cells, the ventricular population in each chamber consists of subpopulations known as trabecular and compact cardiomyocytes[3]. Trabecular cardiomyocytes are the first to develop and are most abundant during fetal life. These cells develop in close contact with the endocardium and form the finger-like structures that protrude into the lumen of the chambers of the fetal heart. Compact cardiomyocytes develop next to the outer epicardium and give rise to the compact myocardium, the thick outer layer of the ventricular wall that provides the contractile force to pump blood throughout life. This myocardium is formed through a process of compaction that involves a transient proliferative stage required to generate sufficient numbers of compact cardiomyocytes to create the muscle[3]. Studies from model organisms have shown that factors secreted from the epicardium including retinoic acid (RA), Wnt, fibroblast growth factors (FGFs) and insulin-like growth factors (IGFs), play important roles in specification and proliferation of the compact lineage in the early fetal heart[4–7].

Cardiomyocytes that make up the fetal heart are immature and display characteristics that distinguish them from their counterparts in the adult organ. These include automaticity, underdeveloped calcium handling capacity, poor mitochondrial oxidation capacity and a low level of contractile apparatus organization[8,9]. A major transition in the maturation status of the cardiomyocytes occurs at birth as the cells switch their energy production from glycolysis to fatty acid oxidation (FAO) to meet the increased workload of the newborn heart. Along with this switch, cardiomyocytes rapidly increase their mitochondrial mass resulting in enhanced mitochondrial oxidation capacity and exit the cell cycle giving rise to the quiescent population found in the adult organ[10,11]. While the regulation of fetal/postnatal cardiomyocyte maturation is not fully understood, evidence from a number of different studies have shown that signaling through peroxisome proliferator-activated nuclear receptors (PPARs) and responses to hormonal changes and lipids play a role in the metabolic switch and mitochondrial biogenesis in the postnatal heart[12–16].

Studies aimed at generating ventricular cardiomyocytes from hPSCs have investigated different stages of this developmental progression, but none has recapitulated the entire developmental path from specification to maturation and function. With respect to the earliest lineage specification step, we have previously shown that the human ventricular lineage develops from a mesoderm population that is distinct from the one that gives rise to atrial cells, recapitulating the finding in the early embryo that these cell types derive from different progenitors[17]. The cardiomyocyte populations generated in our study as well as in most others published to date are immature and display characteristics similar to cardiomyocytes in the early fetal heart. Strategies aimed at maturing hPSC-derived cardiomyocytes have used electromechanical stimulation as well as exposure to pathway agonists and hormones that promote maturation of the metabolic machinery[18–20]. With respect to the latter approach, a number of different studies have shown that the addition of either lipids, hormones or PPARalpha (PPARa) agonists to the differentiation cultures will induce metabolic changes in the hPSC-derived cardiomyocytes indicating that the in vitro-derived cells can respond to stimuli that promote cardiomyocyte maturation in vivo[20–22]. While encouraging, none of these studies included quantitative approaches to determine the efficiency of maturation at the population level and none used well characterized target cardiomyocyte populations. As a consequence, it is difficult to compare the effect of the stimuli used in the different studies and it is not known if the manipulations described can impact the maturation status of different cardiomyocyte subtypes.

In this study, we followed a staged approach to generate functionally mature compact cardiomyocytes from hPSCs focusing on recapitulating the key events that regulate ventricular development and maturation in vivo. We show that Wnt and IGF signaling specify a compact cardiomyocyte population that in response to transient activation of PPARa signaling and hormonal and FA stimuli undergoes a series of maturation steps that include a switch from glycolysis to FAO. The mature cells are able to use exogenous FAs as an energy source, and show improved sarcomere structure, increased contraction force, increased mitochondrial oxidative capacity and changes in global RNA expression patterns. The metabolically mature ventricular cells initiate a reversion to glycolysis and an increase in lipid accumulation and progression to apoptosis following exposure to stimuli that mimic heart failure. Transplantation studies using the nude rat model revealed that the mature cells generated more mature grafts than the immature cells, suggesting that the maturation status of the transplant population can impact the quality of the graft. When exposed to the combination of maturation factors, atrial cardiomyocytes showed similar patterns of maturation, indicating that the strategy is not chamber specific.

## Results

**Specification of the compact cardiomyocyte lineage.** To be able to model diseases that affect the compact ventricular myocardium and to develop cell based therapies to treat them, we focused our initial studies on identifying the signaling pathways that regulate the generation of ventricular compact cardiomyocytes from hPSCs. As a first step, we characterized the trabecular and compact composition of the day 20 cardiomyocyte population generated with the ventricular differentiation protocol previously established in our lab[17] through single cell RNA sequencing (scRNAseq) analyses (Fig. 1a–c, Supplementary Table 1). T-SNE plots of clustered scRNAseq data identified 9 distinct groups within the $cTNT^+$ cardiac population (Fig. 1b). The large majority of the cells in the different clusters expressed MYL2 confirming that they are ventricular cardiomyocytes (Fig. 1c). Most of the $MYL2^+$ cells expressed HEY2 a marker of the compact fate[23], indicating that a high proportion of the population is compact cardiomyocytes. A small subpopulation (cluster 4) of HEY2 negative cells expressed the trabecular marker NPPA suggesting that the day 20 population also contains some trabecular

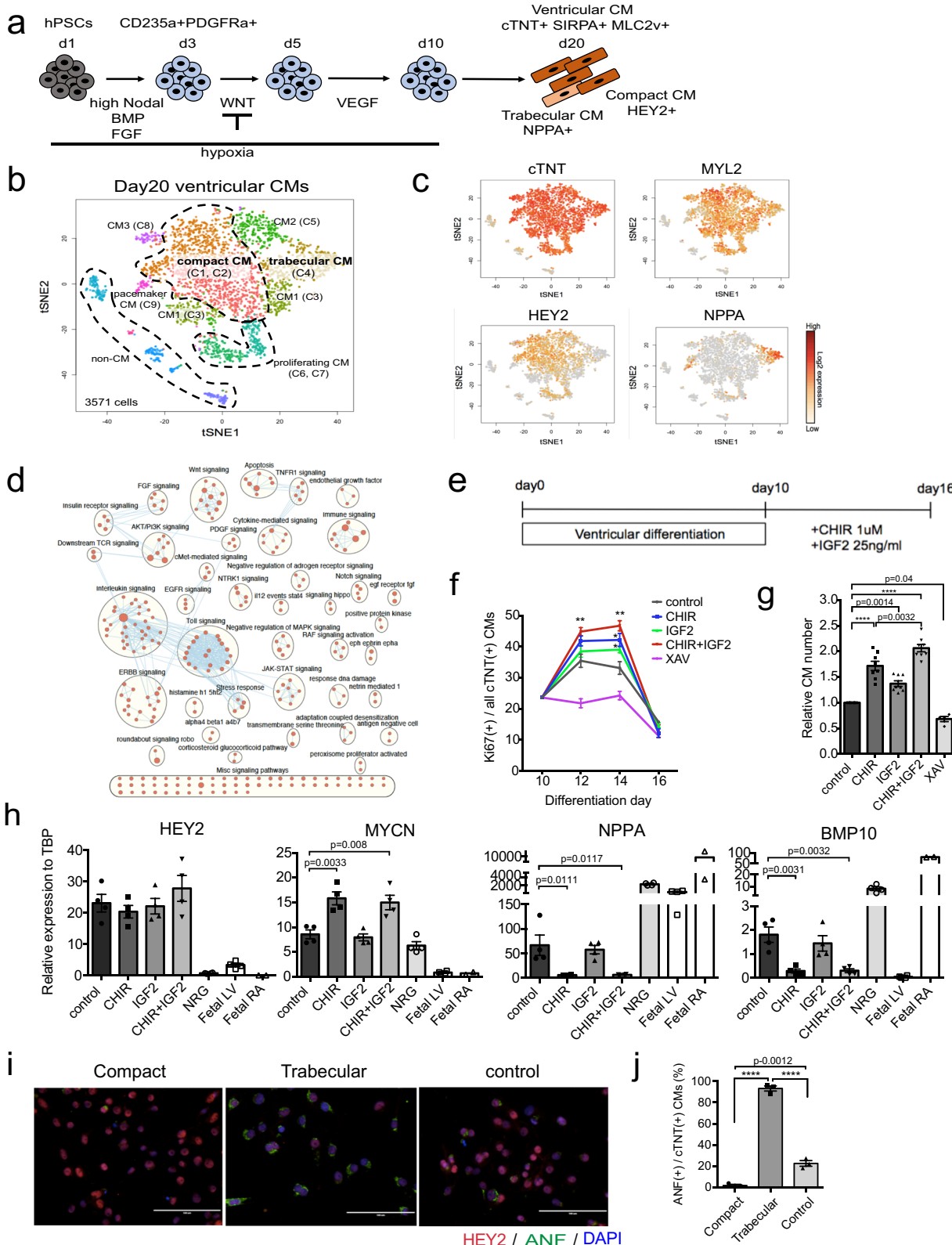

Components of the Netrin 1 and Wnt pathways

cardiomyocytes (Fig. 1c). More detailed analyses of the *HEY2* positive clusters (1 and 2) and the *NPPA* expressing cluster (4) confirmed this lineage assignment (Supplementary Fig. 1a). *HIF1A* and *CCND2*, genes associated with compact myocardium[24,25], were expressed at higher levels in clusters 1 and 2 than in cluster 4 whereas *SCN5A* and *IRX3* known to be preferentially expressed in trabecular myocardium (cluster 4)[26,27] showed the opposite pattern (Supplementary Fig. 1b).

Pathway analyses of clusters 1, 2 and 4 identified a number of signaling pathways that were expressed at higher levels in the compact than in the trabecular cardiomyocytes (Fig. 1d, Supplementary Fig. 1c). Components of the Netrin 1 and Wnt pathways

**Fig. 1 Generation of compact cardiomyocytes. a** Schema of the protocol for ventricular cardiomyocyte differentiation from hPSCs. **b** tSNE plot of a day 20 ventricular cardiomyocyte population showing 9 different clusters in the cTNT[+] subpopulation and **c** the expression patterns of cTNT, MYL2, HEY2 and NPPA within them. **d** Results from pathway analysis showing signaling pathways upregulated in the HEY2[high] cells compared to the NPPA[high] cells.
**e** Protocol for generation of compact cardiomyocytes in vitro. **f** Quantification of the changes in the percentage of Ki67[+] cardiomyocytes in day 10 and 16 cardiomyocyte populations treated as indicated. CHIR (1 uM), IGF2 (25 ng/ml), XAV (4 uM) ($N = 4$). day 12: control vs CHIR + IGF2, $p = 0.009$, day 14: control vs CHIR; $p = 0.0212$, control vs IGF2; $p = 0.0410$, control vs CHIR + IGF2; $p = 0.0021$ by two-sided unpaired $t$-test (*$p < 0.05$, **$p < 0.01$).
**g** Quantification of the relative number of cardiomyocytes in the day 16 populations treated as indicated. DMSO control set as 1 ($N = 8$ for control, CHIR-IGF2-, and CHIR + IGF2-treated, $N = 4$ for XAV-treated). **h** RT-qPCR expression analyses of compact (HEY2 and MYCN) and trabecular (NPPA and BMP10) markers in the indicated populations ($N = 4$). Fetal LV and RA tissues (gestation week 17) were included as a reference for in vivo expression.
**i** Representative immunostaining of compact (CHIR + IGF2 treated), trabecular (NRG treated), and DMSO treated (control) cardiomyocytes. Scale bar: 100 um. **j** Quantification of the percentage of ANF[+]/cTNT[+] cardiomyocytes in compact, trabecular, and non-treated ventricular (control) populations ($N = 3$). Statistical analyses in **g–j** were performed by one-way ANOVA with Tukey's multiple comparisons (****$p < 0.0001$). Data are presented as mean values ± SEM. Fetal LV: Fetal left ventricular tissue ($N = 4$), Fetal RA: Fetal right atrial tissue ($N = 2$). All values shown for the PCR analyses are relative to the housekeeping gene TBP.

were among the most differentially expressed (Supplementary Fig. 1c). Given that Wnt ligands secreted by the epicardium are known to induce proliferation of compact cardiomyocytes in vivo[4,25], we focused our efforts on investigating the role of this pathway in compact cardiomyocyte development from hPSCs. Additionally, as it has been reported that epicardial-derived IGF2 also induces compact layer proliferation in the developing heart[7], we evaluated the role of this pathway in parallel. Using Ki67 as a measure of proliferation, we found that the highest proportion of Ki67[+]cTNT[+] cardiomyocytes was detected between days 12 and 14 of culture in the absence of any added cytokines (Supplementary Fig. 1d). Using this timeframe as an indication of the proliferative stage of ventricular development, we investigated the consequences of adding either the small molecule inhibitor of GSK3, CHIR (Wnt pathway agonist), IGF2 or both to the cultures between days 10 and 16 of differentiation (Fig. 1e). For comparison, we also treated cells with Neuregulin 1 (NRG1) to promote a trabecular fate[28,29]. The addition of either CHIR or IGF2 led to a significant increase in the proportion of Ki67[+]cTNT[+] cardiomyocytes as well as the total number of cardiomyocytes generated (Fig. 1f, g). The induction of Ki67[+] cardiomyocytes and the increase in cell number (2-fold increase) were greatest when both factors were added together. Although the factors were maintained in the cultures until day 16 of differentiation, the proportion of Ki67[+] cells dropped sharply between days 14 and 16, suggesting that the population becomes refractory to the proliferative effects of these cytokines. The addition of NRG1 during this timeframe led to a 1.3-fold increase in cell numbers. Inhibition of Wnt (XAV) in the control cultures (without CHIR or IGF2) led to a reduction in the proportion of Ki67[+]cells suggesting that proliferation in the absence of exogenous stimuli is mediated through endogenous Wnt signaling (Fig. 1f, g). RT-qPCR analyses revealed that Wnt signaling was the main driver of the compact fate as demonstrated by the upregulation of expression of the compact marker MYCN and downregulation of the trabecular markers NPPA and BMP10 in the CHIR-treated population (Fig. 1h). Other compact markers including HEY2, TBX20 and FZD1 that were expressed at high levels in the pre-treated population were not impacted by the addition of CHIR and IGF2 (Fig. 1h, Supplementary Fig. 1e). In contrast to the effects of CHIR, the addition of NRG1 led to a downregulation of HEY2, TBX20 and FZD1 expression and an upregulation of the trabecular markers NPPA, BMP10, IRX3, NPPB and HAS2. Both populations expressed markers indicative of ventricular cardiomyocytes including cTNT, IRX4 and MYH6 (Fig. 1h, Supplementary Fig. 1e).

Immunostaining analyses confirmed the molecular profiles and showed that the untreated day 16 ventricular population was made up of a majority of HEY[+] compact cardiomyocytes and a

small fraction (~20%) of ANF (NPPA)[+] trabecular cells (Fig. 1i, j, Supplementary Fig. 1f). The proportion of ANF[+] cells was significantly reduced following treatment with CHIR/IGF2, indicating that activation of the Wnt/IGF2 pathways specified a population highly enriched in HEY2[+] compact cardiomyocytes. By contrast, treatment with NRG1 efficiently promoted a trabecular fate as demonstrated by the presence of ANF[+] cells and a lack of HEY[+] compact cells (Fig. 1i, j, Supplementary Fig. 1f). Collectively, these findings show that it is possible to efficiently specify compact and trabecular fates from an immature ventricular cardiomyocyte population through staged (days 10–16) manipulation of the Wnt and NRG pathways.

**Induction of fatty acid oxidation in compact cardiomyocytes.** With access to enriched populations of compact cardiomyocytes, we were next interested in identifying the regulatory pathways that promote their maturation, reasoning that more mature cells may function better following transplantation into the adult infarcted heart and provide the appropriate target population for modeling cardiovascular disease in vitro. Following their proliferative stage, compact cardiomyocytes undergo a series of maturation steps that include a reduction in proliferation and a shift in energy metabolism from glycolysis to fatty acid oxidation (FAO). To promote the exit from cell cycle, we treated the developing cardiomyocyte population with the WNT inhibitor XAV for 2 days given that our previous analyses showed that endogenous Wnt signaling can promote proliferation. As shown in Supplementary Fig. 2a, the addition of XAV following the CHIR/IGF2-induced proliferative stage significantly reduced the proportion of Ki67[+] cells in the population.

To measure the competence of the XAV-treated population to undergo FAO, we first analyzed the day 18 compact cardiomyocytes for expression of the FA transporter CD36, essential for the efficient transport of FAs into cells[30] (Supplementary Fig. 2b). As shown in Fig. 2a, very few cells within the day 18 population expressed CD36 consistent with the interpretation that they are metabolically immature. A small CD36[+] population emerged following an additional 2 weeks of culture (day 32), suggesting that the cells are undergoing the initial stages of the metabolic transition to FAO in the absence of additional manipulations (Fig. 2a). To promote the metabolic switch in cardiomyocytes, we evaluated the effects of PPAR-related signaling, the response to steroid (dexamethasone) and thyroid (T3) hormones and the response to FA (palmitate), all of which have been shown to regulate FAO and mitochondrial function[12,15,31,32]. Palmitate is a common dietary fatty acid that can be incorporated into multiple fatty acid synthesis and oxidation pathways. RT-qPCR analyses showed that the day 10 and day 18 cardiomyocytes expressed PPARa as well as the thyroid hormone (THRA) and glucocorticoid receptors

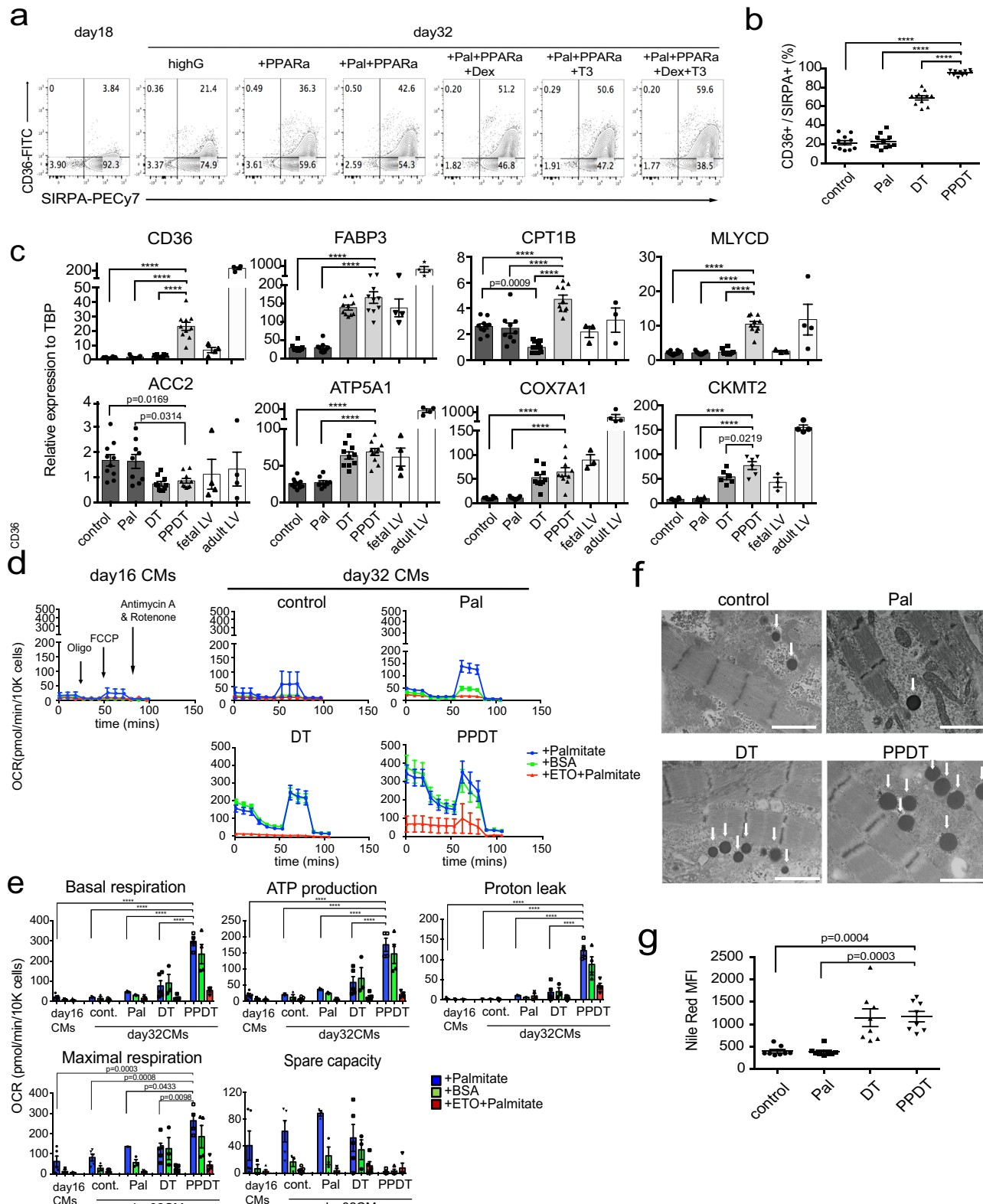

(NR3C1) suggesting that they should respond to these stimuli (Supplementary Fig. 2c). Addition of the PPARa agonist (GW7647, 1 uM) to day 18 cardiomyocytes led to a significant increase in the size of CD36+ population detected at day 32 of culture (Fig. 2a, Supplementary Fig. 2d). The size of CD36+ population increased further following the addition of palmitate (Pal) to the PPARa agonist-treated cultures (Fig. 2a, Supplementary Fig. 2d). Neither

dexamethasone (Dex) nor thyroid hormone (T3) added alone with the PPARa agonist and palmitate impacted the size of the CD36+ population. However, the addition of both hormones with the PPARa agonist and palmitate did promote the development of a significantly larger CD36+ subpopulation, which represented ~50% of the total day 32 SIRPA+ cardiomyocyte population (Fig. 2a, Supplementary Fig. 2e).

**Fig. 2 Induction of FAO in compact cardiomyocytes. a** Representative flow cytometric analyses of CD36 and SIRPA expression on day 18 and derivative day 32 cardiomyocyte populations cultured in the indicated conditions; GW7647 (PPARa agonist; 1 uM), Palmitate (200 uM), Dexamethasone (Dex) (100 ng/ml), T3 (4 nM). **b** Quantification of the proportion of CD36$^+$/SIRPA$^+$ cells in day 32 populations cultured from day 18 to 32 in the indicated conditions ($N = 11$). **c** RT-qPCR expression analyses of FAO-related genes in day 32 cardiomyocytes cultured in the indicated conditions ($N = 10$ for control, DT-, and PPDT-treated, $N = 9$ for Pal-treated for analysis of CD36, FABP3, CPT1B, MLYCD, ACC2, ATP5A1, and COX7A1, $N = 6$ for all conditions for CKMT2 analyses). Fetal LV and adult LV tissues were included as a reference for in vivo expression. **d** Representative kinetics of the oxygen consumption rate (OCR) measured with FAO Cell Mito stress test assay in day 16 immature cardiomyocytes (day 16 immature) and day 32 cardiomyocytes cultured in the indicated conditions. (day 16; data are from 2 technical replicates in palmitate, single sample in BSA, single sample in ETO, day 32 control; data from 2 technical replicates in palmitate, single sample in BSA, single sample in ETO, day 32 Pal; data are from 9 technical replicates in palmitate, 10 technical replicates in BSA, 2 technical replicates in ETO, day 32 DT; data are from 10 technical replicates in palmitate, 10 technical replicates in BSA, 2 technical replicates in ETO, day 32 PPDT; data are from 9 technical replicates in palmitate, 10 technical replicates in BSA, 2 technical replicates in ETO). **e** Comparison of each parameter of the FAO Cell Mito stress assay in the cardiomyocytes cultured under the indicated conditions (day 16; $N = 5$, day 32 control; $N = 5$, day 32 Pal; $N = 3$, day 32 DT; $N = 5$, day 32 PPDT; $N = 4$ biologically independent experiments). **f** Representative transmission electron microscope images of lipid droplets (white arrows) in cardiomyocytes cultured under the indicated conditions. Scale bar; 1 um. **g** Flow cytometric analyses of Nile Red staining in day 32 cardiomyocytes treated as indicated ($N = 8$). All statistical analyses were performed by one-way ANOVA with Tukey's multiple comparisons (****$p < 0.0001$). Data are presented as mean values ± SEM. Fetal LV: Fetal left ventricular tissue ($N = 3-4$), Adult LV: Adult left ventricular tissue ($N = 3-4$).

As the above studies were carried out in the presence of high concentrations of glucose (4.5 g/L), we next investigated the effects of reducing or eliminating glucose from the media in an effort to further increase the generation of CD36$^+$ cells and promote the use of FAs. Elimination of glucose from our standard 24 well cultures resulted in embryoid body (EB) clumping and massive cardiomyocyte death (Supplementary Fig. 2f). To overcome this problem, we expanded the culture format to larger dishes (10 cm) that could be rotated (from day 10 onwards) to maintain small uniform EBs. Under these conditions, the cells survived well in the presence of high (4.5 g/L) and low (2 g/L) concentrations of glucose (Supplementary Fig. 2g). Cardiomyocytes also survived in the absence of glucose, although significant cell death (30%) was still observed. In addition to improved cell survival, the modified culture format supported the efficient induction of CD36$^+$ cells (>90%) following treatment with the combination of the PPARa agonist, palmitate, Dex and T3 (PPDT) in the presence of low glucose (Supplementary Fig. 2h, i).

With these optimized conditions, we next compared the efficiency of CD36 induction with PPDT to factors previously reported to induce cardiomyocyte maturation including FAs in low glucose media (palmitate, Pal) and the combination of Dex and T3 in high glucose media (DT) (Fig. 2b)[21,22,33]. Under these conditions, Pal alone did not promote the development of CD36$^+$ cells. The combination of Dex and T3 (DT) induced a sizeable CD36$^+$ population, however these hormones were not as effective as the combination of PPDT which consistently generated cardiomyocyte populations of greater than 90% CD36$^+$ cells (Fig. 2b).

Upon entry into the cell, FAs are shuttled through the cytoplasm to the mitochondria by FABPs and then into the mitochondria by the transporter CPT1. CPT1 mediated transport is inhibited by malonyl CoA, the levels of which are regulated by MLYCD, an enzyme that converts malonyl CoA to acetyl CoA, and ACC2, that converts acetyl CoA to malonyl CoA (Supplementary Fig. 2b). RT-qPCR analyses revealed that treatment with PPDT led to significant increases in the expression levels of the above components of the FAO pathways, including *CD36*, *FABP3*, *CPT1B*, and *MLYCD*, over those detected in untreated populations (Fig. 2c). The level of *ACC2* by contrast was significantly downregulated. We also observed an upregulation of expression of genes that encode components of mitochondrial function and the electron transport chain, including *ATP5A1*, *COX7A1*, and *CKMT2* (Fig. 2c). Treatment with Pal alone had no effect on the expression patterns of any of these genes whereas addition of DT induced the upregulation of *FABP3*, *ATP5A1*,

*COX7A1* and *CKMT2*. The expression of *CPT1B* and *MLYCD* was only upregulated following treatment with PPDT. These observations are consistent with the CD36 flow cytometric analyses and indicate that the combination of PPDT is the most effective at initiating the metabolic switch from glycolysis to FAO.

To functionally assess the capacity of the different populations to import and metabolize exogenous FAs, we used the Seahorse XF assay (FAO Cell Mito Stress Assay) to measure oxygen consumption rate (OCR) in the presence of palmitate (blue line) or BSA as a control (green line). Quantification of the OCR parameters revealed that the PPDT-treated cells demonstrated higher basal metabolism, ATP production, proton leak and maximal respiration capacity than untreated cells or those treated with Pal or DT (Fig. 2d, e). Addition of etomoxir (ETO, red line), an inhibitor of FAO (inhibitor of CPT1), blocked mitochondrial oxidation demonstrating that the observed OCR is dependent on FAO. Although treatment with PPDT initiated the switch to FAO, no significant differences were detected in the OCR in the presence of Palmitate (blue line) compared to BSA (green line), suggesting that the cells are relying on an endogenous FA source (Fig. 2d, e). Similar patterns were observed in the DT-treated cells, indicating that the hormonal stimuli promote the use of endogenous FA rather than exogenous FA. The PPDT-treated cells showed no spare energy capacity demonstrating that, at a basal level, they were oxidizing substrates at maximal capacity (Fig. 2d, e). Additionally, these cells also had a significant amount of proton leak, known to be activated by increased rates of oxidative phosphorylation[34]. This increase in oxidative capacity leads to higher amounts of free protons and reactive oxygen species (ROS) in the inner mitochondrial space resulting in oxidative stress and further proton leak[35]. To compensate for this increased oxidative stress, cells upregulate expression of uncoupling protein (UCP) and anti-oxidative stress (SOD2) genes[36,37]. RT-qPCR analyses showed that the PPDT-treated cells expressed higher levels of *UCP2* than the other populations and higher levels of *SOD2* than the control and Pal-treated cells (Supplementary Fig. 2j).

The apparent use of endogenous FA as an energy source may be reflective of the pattern of FA usage in the neonatal heart, as the cells are transitioning from glycolysis to FAO. Immediately after birth, lipid droplets accumulate within the cardiomyocytes in the heart, likely serving as an endogenous lipid reserve as the cells undergo the switch to FAO[10]. Transmission electron microscopy (TEM) analyses of the PPDT- and DT-treated cells showed that both populations contained structures resembling lipid droplets (Fig. 2f). Quantification of the lipid droplet

membrane area measured by Nile Red staining revealed that these populations contained significantly larger positive areas than the untreated or Pal-treated cells (Fig. 2g). The presence of lipid droplets in these cells would support the interpretation that they have access to internal stores of FA and as such would provide an explanation as to why we did not observed differences in the OCR in the presence of palmitate (blue line) or BSA (green line).

Collectively, these findings indicate that the hPSC-derived mature compact cardiomyocytes undergo changes associated with the switch to FAO observed in the newborn heart, including the metabolism of long chain FAs, the storage of lipids and the upregulation of anti-oxidative stress genes to protect against the effects of oxidative stress.

**Transient activation of FAO improves metabolic profiles**. The lack of spare capacity in the PPDT-treated population suggests that the prolonged maturation stimulus may be inducing an abnormal hyperactive stressed phenotype in these cells. To determine if manipulation of the duration of treatment could impact metabolic function, we shortened the induction time from 2 weeks to 9 days and then maintained the cells in either Pal and PPARa, Pal alone or no factors for the remaining 5 days (Fig. 3a). These conditions were designed to mimic transient activation of the FAO pathway and the hormonal surge observed in the neonatal heart during the early postnatal adaptation period[16,38,39]. Seahorse analyses revealed that cells maintained in Pal alone for the final 5 days of culture showed significant increase in maximal respiration and spare capacity (Fig. 3b3). Notably, we now observed a large difference in OCR between cells in the presence of palmitate (blue line) and those in the presence of BSA (green line), indicating that the cells are competent to use an exogenous source of lipids for energy. The increase in spare capacity and the ability to use exogenous lipids was lost with the removal of palmitate, suggesting that the presence of long chain FAs is essential for maintaining the FAO state of the cells (Fig. 3b4). Quantification of the OCR curve parameters showed that cells treated for 9 days with PPDT and then cultured in Pal alone showed significantly higher maximal respiration and spare capacity compared to cells treated with the other combinations of factors (Fig. 3c). Nile Red staining revealed that the shortened induction time led to a reduction in the endogenous lipid stores in the cells compared to those subjected to continuous activation (day 18–32) (Fig. 3d). RT-qPCR analyses showed some differences between the populations subjected to continuous stimulus and those induced for 9 days followed by culture in Pal (transient) including a reduction in the expression levels of *CD36*, *CPT1B*, *MLYCD* and *UCP2*, genes involved in FAO (Supplementary Fig. 3). In contrast, the levels of mitochondrial related genes, such as *ATP5A1*, *COX7A1*, and *CKMT2*, were similar in the two populations (Supplementary Fig. 3). Together, these findings demonstrate that induction of hPSC-derived cardiomyocytes for 9 days with PPDT followed by 5 days of culture in Pal (PPDT/PAL) (Fig. 3e) promotes metabolic maturation of the cells yielding a population that displays a metabolic phenotype similar to that of postnatal cardiomyocytes including the ability to oxidize exogenous FAs. We will refer to these cardiomyocytes as 'mature' in the following studies.

**Structural and functional changes in mature cardiomyocytes**. Along with changes in metabolism, the mature cells showed differences in structural and functional properties compared to the immature cells. The mature cells were larger than those in the other groups and the population contained more bi-nucleated cells than the control or Pal treated populations (Fig. 4a–c). Transmission electron microscope (TEM) analyses showed that the mature cardiomyocytes had longer sarcomeres with more organized structure including detectable Z lines, I bands, and A bands and larger mitochondria with more defined cristae matrix than the control cells or those treated with the other combinations of factors (Fig. 4d–g, Supplementary Fig. 4a). $Ca^{2+}$ transient analyses in the monolayer format using Fluo4 dye revealed that the mature cardiomyocytes as well as those treated with DT displayed improved $Ca^{2+}$ handling capacity compared to the untreated control and the Pal-treated cardiomyocytes (Fig. 4h, i). As mature cardiomyocytes are quiescent, we next measured the proportion of Ki67$^+$ cells in each population as an additional indication of maturation status. These analyses showed that the percentage of Ki67$^+$cTNT$^+$ cardiomyocytes was significantly lower in mature and DT-treated populations than in the control or Pal-treated populations (Supplementary Fig. 4b). To determine if the day 32 populations contained cells that could still respond to proliferative signals, each was treated with CHIR and IGF2 for 2 days and then analyzed for the presence of Ki67$^+$ cells (Supplementary Fig. 4c, d). As shown in Supplementary Fig. 4d, the untreated control, as well as the Pal- and DT-treated populations contained CHIR/IGF2 responsive cells. In contrast, no significant response was detected in the mature population, suggesting that these cells have lost their capacity to respond to these proliferative stimuli (Supplementary Fig. 4d).

Given the observed structural differences between the cells in the various populations, we next evaluated their contraction force using engineered 'biowire' cardiac tissues[40]. For these studies, tissues generated with day 18 compact cardiomyocytes were treated with the different combinations of factors (no factors, Pal, DT or PPDT/PAL) for two weeks and then analyzed for contraction force as previously described[41]. As shown in Fig. 4j, the contraction force of the PPDT/PAL-treated (mature) tissue was significantly higher than the non-treated control tissues or those treated with Pal or DT (Fig. 4j). TEM analyses showed that the cells in the mature tissues had more distinct, mature sarcomere structures than the cells in the tissues of the other groups (Supplementary Fig. 4e). Immunocytochemical analyses revealed the presence of comparable proportions of cTNT$^+$ cardiomyocytes and CD90$^+$ fibroblasts in the tissue constructs indicating that the differences in force are not due to dramatic differences in the proportion of these cell types (Supplementary Fig. 4f, g).

Collectively, the findings from this set of analyses show that the PPDT/PAL-treated cells display features of maturation that extend beyond a switch in metabolism and include changes in cell morphology, calcium handling capacity and force contraction.

To understand the role of FAO in cardiomyocyte maturation, we next tested the effect of inhibiting this metabolic pathway through the addition of etomoxir (ETO) to the cultures during the maturation process (from day 18 to day 32). RT-qPCR analyses of treated populations showed that the expression levels of the mitochondrial genes *CKMT2*, *COX7A1*, and *COX6A2*, the sarcomere genes *MYL2* and *MYOZ2* and the $Ca^{2+}$ handling related gene *ATP2A2* were significantly lower in the ETO-treated cells than in the control population (Fig. 4k), suggesting that FAO does promote aspects of cardiomyocyte maturation. However, not all the molecular changes associated with maturation were inhibited by the addition of ETO as the expression levels of other sarcomere (*TCAP*, *DES*, *MYOM3*), mitochondria (*ATP5A1*, *ATP1A3*), $Ca^{2+}$ and ion channel (*KCNJ2*, *HRC*, *CALM1*)-related genes were not affected, indicating that the combination of PPDT promotes aspects of maturation independent of the switch to FAO.

**Molecular profiling of mature cardiomyocytes**. To further characterize the mature compact cardiomyocytes, we carried out

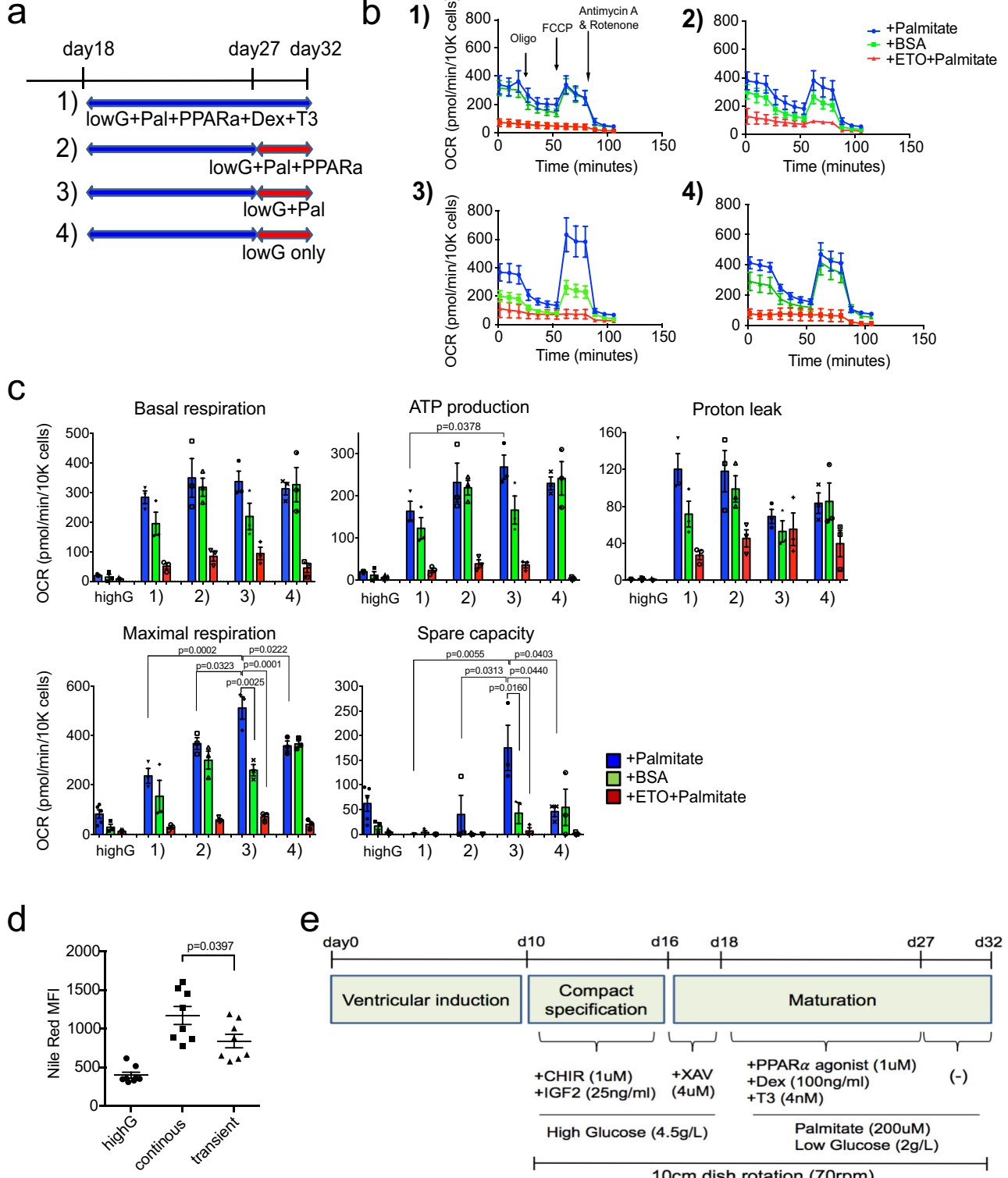

scRNAseq analysis on the day 32 mature population and compared these profiles to those from the untreated age matched immature population (Supplementary Table 1). The mature and immature populations used for these analyses consisted of 92% and 85% cTNT[+]/MLC2V[+] cells respectively (Supplementary Fig. 5a). UMAP analyses of the combined data set identified 5 distinct clusters, three of which expressed high levels of *TNNT2* (*cTNT*) (clusters 0, 1, and 3), one enriched for extracellular matrix (ECM)-related genes (cluster 2) and one that expressed

endoderm-related genes (cluster 4) (Fig. 5a, b, Supplementary Table 2). In addition to *TNNT2* (*cTNT*), Cluster 3 also expressed smooth muscle-related genes (Fig. 5b, Supplementary Table 2). Cluster 0 expressed higher levels of genes associated with FAO (*CD36, FABP3, ACSL1*) and mitochondrial (*CKMT2, COX6A2*) and muscle function (*DES, MYBPC3, ACTN2*) than cluster 1 (Fig. 5c), confirming that cluster 0 represents the mature cardiomyocytes while cluster 1 contains the immature cardiomyocytes (Fig. 5d).

**Fig. 3 Transient activation of the FAO pathway improves metabolic profiles in mature compact cardiomyocytes. a** Experimental scheme used to determine the effects of manipulating the duration of PPDT treatment. Cardiomyocytes were treated either for the full 14 days with PPDT or for 9 days with PPDT followed by culture for 5 days in the indicated conditions. **b** Representative kinetics of OCR in the four conditions depicted in **a** 1); data are from 7 technical replicates in palmitate, 9 technical replicates in BSA, 2 technical replicates in ETO, 2); data are from 7 technical replicates in palmitate, 5 technical replicates in BSA, 2 technical replicates in ETO, 3); data are from 9 technical replicates in palmitate, 8 technical replicates in BSA, 2 technical replicates in ETO, 4); data are from 6 technical replicates in palmitate, 7 technical replicates in BSA, 2 technical replicates in ETO). **c** Comparison of each of the parameters measured in the FAO Cell Mito stress assay in day 32 cardiomyocytes cultured in the condition shown in **a** ($N = 3$ biologically independent experiments). Statistical analyses were performed by one-way ANOVA with Tukey's multiple comparisons. **d** Flow cytometric analyses of Nile Red staining in day 32 cardiomyocytes treated either continuously with PPDT (continuous) or for 9 days followed by 5 days of culture in Pal (transient) ($N = 8$). Statistical analyses were performed by two-sided unpaired $t$-test. **e** Protocol for the generation of mature compact cardiomyocytes. Data are presented as mean values ± SEM.

To elucidate differences between the cardiomyocytes in cluster 0 and 1, we performed Gene Ontology (GO) enrichment analysis using genes differentially expressed between these 2 clusters. This analyses showed that the mature cells (cluster 0) expressed higher levels of genes associated with FAO, mitochondrial function, muscle contraction and sarcomere organization than the immature cardiomyocytes (cluster 1), further confirming that the PPDT/PAL treatment induces broad maturation changes within the cells (Fig. 5e). Analyses of a curated list of well characterized genes from these different groups showed higher levels of expression of most in the mature compare to the immature cells (Supplementary Fig. 5b). Several notable exceptions were observed. The first is in the switch in the TNNI1/3 isoforms known to occur during cardiomyocyte maturation[42]. While our mature cells showed lower levels of the fetal form *TNNI1* than the immature cells, the levels of the adult isoform, *TNNI3* were comparable in the 2 populations. The reason for this is currently not clear, as the mature cells have upregulated many other sarcomere genes. The second is in the patterns of *MYH6* and *MYH7* expression that are opposite of what one would expect in mature and immature cells. This reverse pattern may be due to the presence of T3 in our maturation cocktail as this hormone has been shown to induce *MYH6* and inhibit *MYH7* expression[19,43].

To define a molecular signature for the metabolically mature cells, we identified genes whose expression patterns are highly discriminatory (on/off) between the populations. This list of genes includes *CD36* as well as several lipid metabolism genes (*NMRK2, ACSL1, SCD, MASP1, TP53INP2*) and adipogenesis-related transcription factors (*KLF9, CEBPB*)[44] (Fig. 5f, asterisk*). Gene Set Enrichment Analysis (GSEA) between the two populations revealed that the TGACCTTG regulatory motif, which is reported to be a binding site of ESRRA (Estrogen-related receptor A)[45], was highly enriched and overlapped with FA metabolism and oxidative phosphorylation genes in the mature cells (Fig. 5g, Supplementary Fig. 5c). Consistent with these findings, expression analysis showed that the mature cardiomyocytes had higher levels of *ESRRA* than the immature cells (Fig. 5h). In contrast, the expression levels of *ESRRG*, another member of this receptor family were lower in the mature cells (Supplementary Fig. 5d).

More detailed analyses of the mature cardiomyocyte population (cluster 0) revealed heterogeneity that resolved into 4 distinct clusters (Fig. 6a, Supplementary Table 3). The major distinguishing features of these subpopulations were the expression of stress-related genes, *ATF5* and *TRIB3* in cluster B, genes indicative of proliferation (*MKI67* and *FOXM1*) in cluster C and extracellular matrix related genes such as *FN1* and *COL3A1* in cluster D (Supplementary Fig. 6a). These findings indicate that the mature population consists of a large subpopulation of mature, non-proliferating, non-stressed cardiomyocytes (cluster A), along with a subpopulation of proliferating cells (cluster C), a small subpopulation of contaminating fibroblasts (cluster D) and a

subpopulation of stressed cells (cluster B). GO analysis using differentially expressed genes in each cluster revealed that muscle stress fiber-related genes and cholesterol import-related genes were upregulated in cluster A, while ER stress-related genes including those of the CHOP-C/EBP complex and the CHOP-ATF4 complex were upregulated in cluster B (Fig. 6b). Further analyses identified a number of genes that were expressed at higher levels in cluster A than in the other clusters including the surface markers *CD36* and *LDLR* (Fig. 6c), the cytokine *FGF12*, the enzyme *ASB2*, as well as those annotated to contraction, $Ca^{2+}$ handling, extracellular matrix production, metabolism, and several with unknown function in the context of cardiomyocytes (Supplementary Fig. 6b, Supplementary Table 3). To confirm the differences in LDLR at the protein level, we used flow cytometric analyses to monitor its cell surface expression and compared it to that of CD36 (Fig. 6d, Supplementary Fig. 6c). Neither marker was detected at day 18 and only small subpopulations of positive cells were present in the day 32 immature population. In contrast, more than 50% of the mature population expressed both markers at day 32. $CD36^+LDLR^+$ cardiomyocytes were already detected at day 25 at which point they represented ~35% of the maturing population. Taken together, the findings from these flow cytometric analyses show that expression of CD36 and LDLR are upregulated as the ventricular cardiomyocytes undergo metabolic maturation and as such are ideal markers to monitor these changes.

Our observation that *ESRRA* expression is upregulated in the mature cardiomyocytes is in line with a recent study showing that this receptor along with ESRRG functions to regulate postnatal cardiac maturation in the mouse[46]. Given this, we were next interested in determining if ESRRA plays a role in the maturation of the hPSC-derived cardiomyocytes under the conditions used here. To address this, we first tested the different components of our maturation cocktail to determine which was responsible for inducing ESRRA expression. As shown in Fig. 6e, the combination of Dex and T3 appears to be the main driver of ESRRA expression in the mature cardiomyocytes. PPARa agonist did not induced ESRRA expression, highlighting important differences in the effect of these stimuli. We next performed siRNA-mediated knockdown of ESRRA to determine if it plays any role in the maturation of the human cardiomyocytes. For these studies, a mixture of three different siRNAs specific for ESRRA (siESRRA) or a non-targeting siRNA were introduced into day 18 cells. The treated cells were aggregated for 2 days and the aggregates were then cultured for 14 days in our maturation cocktail. Treatment with the siESRRA reduced the levels of *ESRRA* expression to those observed in the immature population (Fig. 6f). This inhibition of ESRRA expression resulted in lower levels of expression of a subset of FAO (*FABP3, ACSL1*) and sarcomere (*MYL2, TCAP*)-related genes compared to the control cells (Fig. 6g), indicating that ESRRA is involved in aspects of metabolic and structural maturation. We also observed an

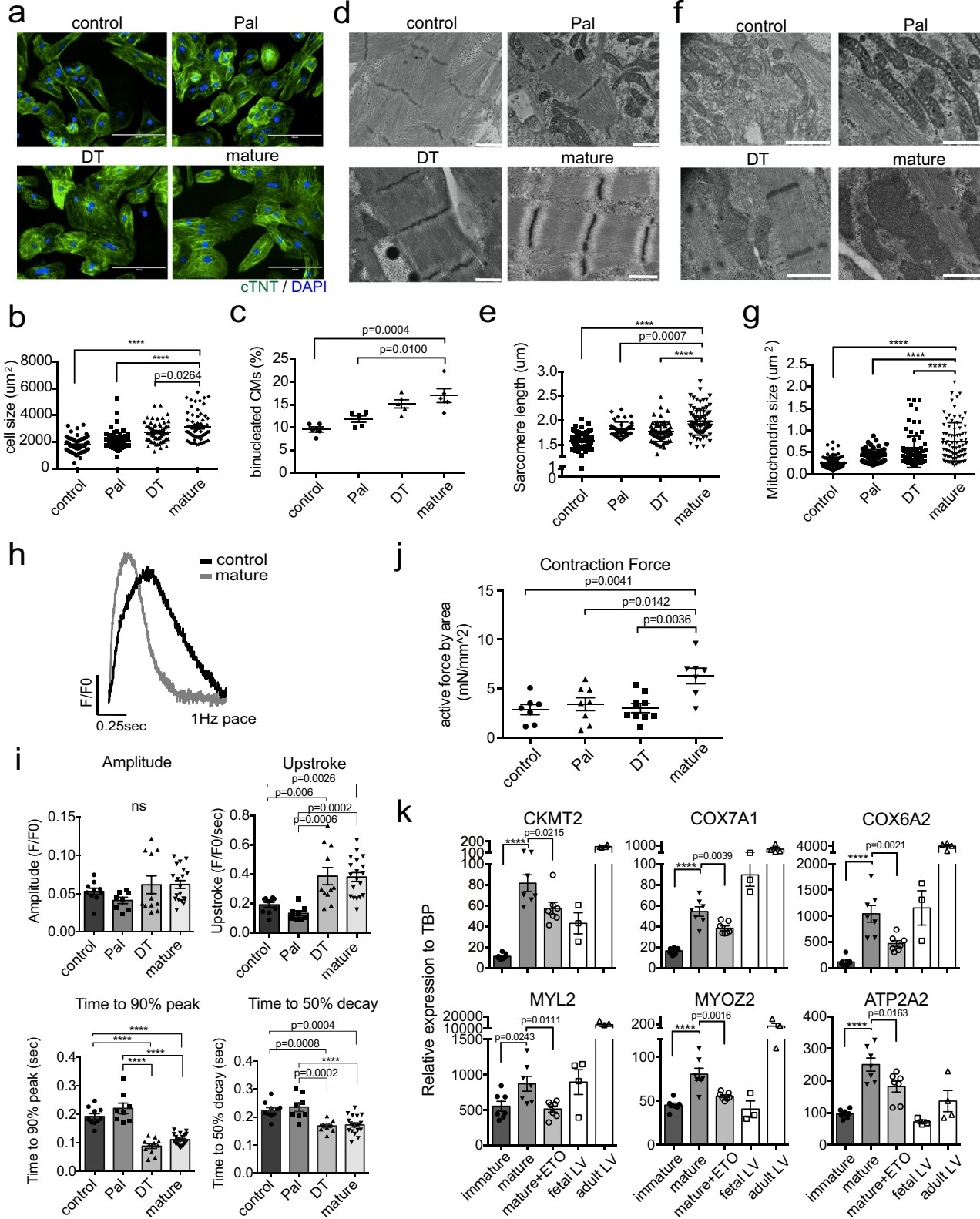

increase in Nile red staining and a downregulation of expression of the hormone sensitive lipase *HSL* that functions to release FAs from lipid droplets in the ESRRA KD cells (Fig. 6h, i). These findings suggest an increase in lipid storage indicating that ESRRA also plays a role in the utilization of endogenous lipids. While these differences were detected, the levels of many other genes including *ESSRG* and *PPARA* (Supplementary Fig. 6d,

Supplementary Table 4) were not impacted by the down-regulation of ESRRA suggesting that this pathway is only part of regulatory machinery that controls cardiomyocyte maturation.

To demonstrate that the protocol described here is applicable across hPSC lines, we used it to promote maturation of cardiomyocytes generated from the hESC line ESI-17. As the efficiency of cardiomyocyte differentiation was not as high as that

**Fig. 4 Structural characteristics of metabolically mature compact cardiomyocytes. a** Representative immunostaining of cardiomyocytes generated under the indicated conditions. Scale bar: 100 um. **b** Quantification of the size of the cardiomyocytes ($N = 73$ cells from control, 67 cells from Pal, 66 cells from DT, and 66 cells from mature conditions examined over 5 independent experiments) and **c** the proportion of binucleated cardiomyocytes generated under the indicated conditions ($N = 5$ independent experiments). **d** Representative transmission electron microscope (TEM) images of sarcomere structures (Scale bar; 1 um) and **e** Quantification of sarcomere length (based on TEM analyses) in the cardiomyocytes generated under the indicated conditions ($N = 72$ cells from control, 43 cells from Pal, 67 cells from DT, and 78 cells from mature conditions examined over 3 independent experiments). **f** Representative TEM images of mitochondria structure (Scale bar; 1 um) and **g** Quantification of mitochondria size in cardiomyocytes generated under the indicated conditions ($N = 76$ cells from control, 64 cells from Pal, 90 cells from DT, and 84 cells from mature conditions examined over 3 independent experiments). **h** Representative $Ca^{2+}$ transient in day 32 control and mature cardiomyocytes. **i** Quantification of the parameters associated with $Ca^{2+}$ transient measurement ($N = 10$ samples from control, 8 samples from Pal, 11 samples from DT, and 19 samples from mature conditions examined over 3 independent experiments) and **j** Quantification of contraction force of biowire cardiac tissues generated under the indicated conditions ($N = 7$ tissues from control, 8 from Pal, 9 from DT, and 7 from mature conditions. All plots were derived from biologically independent experiments). **k** RT-qPCR expression analyses of mitochondria-related genes (*CKMT2, COX7A1, COX6A2*), sarcomere genes (*MYL2, MYOZ2*), and $Ca^{2+}$ handling gene (*ATP2A2*) in immature and mature cardiomyocytes and mature cardiomyocytes cultured with etomoxir (40 uM) (mature+ETO) ($N = 7$). Fetal LV and adult LV tissues were included as a reference for in vivo expression. All statistical analyses were performed by one-way ANOVA with Tukey's multiple comparisons (****$p < 0.0001$). Data are presented as mean values ± SEM. Fetal LV: Fetal left ventricular tissue ($N = 3-4$), Adult LV: Adult left ventricular tissue ($N = 4$).

of the HES2 line, we isolated the SIRPA+ cells from the day 32 mature and immature populations for the molecular analyses (Supplementary Fig. 7a, b). Flow cytometric analysis showed that the ESI-17 derived mature cardiomyocytes showed a significant upregulation of CD36 and LDLR compared to the immature population (Supplementary Fig. 7c, d). Consistent with these changes, our RT-qPCR analyses revealed that the mature cells expressed significantly higher levels of sarcomere, FAO, mitochondria, $Ca^{2+}$ handling, and ion channel-related genes compared to the immature control cells, while the expression level of *HCN4*, the pacemaker current gene, was lower in the mature cells (Supplementary Fig. 7e). Notably, we found that the expression levels of our maturation signature genes (including CD36) were also upregulated in the ESI17-derived mature cardiomyocytes verifying their utility as markers of cardiomyocyte maturation (Supplementary Fig. 7e). These findings clearly show that our maturation protocol can be applied to cardiomyocytes from different cell lines.

**Metabolic maturation of atrial cardiomyocytes.** To determine if the above strategy could be used to promote maturation of other cardiomyocyte subtypes, we treated day 18 atrial cells generated with our previously published protocol[17] with the combination of PPDT/PAL for 14 days. Both the untreated and treated day 32 populations expressed atrial specific genes, including *KCNA5, KCNJ3, GJA5* (CX40) and *NR2F2* (COUPTF2), but no *MYL2* (MLC2V), indicating that they consisted predominantly of atrial cells with few, if any contaminating ventricular cardiomyocytes (Supplementary Fig. 8a, b). TEM analyses showed improved sarcomere structure and increased sarcomere length in the PPDT/PAL-treated (mature) atrial cells compared to the non-treated controls (Fig. 7a, b). PPDT/PAL treatment also led to an increase in mitochondrial size in the atrial cardiomyocytes, although the overall size did not reach that found in the mature ventricular cells ($p < 0.01$ compared to the size in Fig. 4g) (Fig. 7c, d). As observed with the ventricular lineage cells, the PPDT/PAL-treated mature atrial cells also upregulated CD36, however the proportion of positive cells was somewhat lower than observed in the ventricular cardiomyocyte population (Fig. 7e, f). Molecular analyses revealed that PPDT/PAL treatment led to an upregulation of expression of genes associated with the metabolic switch (*CD36, FABP3, CPT1B, MLYCD*) and mitochondrial activity (*ATP5A1, COX7A1, CKMT2*) (Fig. 7g, Supplementary Fig. 8c). The treated cells also showed higher levels of *KCNJ2*, an ion channel gene, and *TCAP*, while *HCN4* showed an opposite pattern and was found at lower levels in the mature than in the immature cells (Supplementary Fig. 8d). OCR measurements

using the Seahorse assay showed that basal respiration, ATP production, proton leak, and maximal respiration were enhanced by the PPDT/PAL treatment of the atrial cardiomyocytes, although the levels of most of these parameters were significantly lower than those in mature ventricular cells (compared to the value in Fig. 3c) (Fig. 7h, i). As observed with the ventricular cells, ETO (red line) blocked the mitochondrial oxidation, indicating that these atrial cardiomyocytes are also dependent on FAO. Together, these findings demonstrate that the factors that regulate maturation of the ventricular lineage cells also promote maturation of hPSC-derived atrial cells.

**Modeling pathological changes in mature cardiomyocytes.** It is well established that heart failure is characterized by distinct changes in myocardial metabolism, including a switch from primarily FA oxidation to glycolysis[47]. These metabolic changes are associated with increased concentrations of FA and lipid accumulation in the form of lipid droplets that, when excessive can lead to cardiomyocyte death[48]. Given that the mature cardiomyocytes induced with PPDT/PAL have acquired the capacity to undergo FAO, they should provide a platform for modeling some of the above metabolic (and lipid accumulation) changes associated with heart failure. As hyperstimulation of the sympathetic adrenergic system is a characteristic of heart failure and activation of the pathway is a well establish pathological stimuli, we analyzed our mature cardiomyocytes for expression of adrenergic receptor B1 (ADRB1), known to bind adrenaline and mediate responses in the adult heart. RT-qPCR analysis showed that the mature cardiomyocytes express significantly higher levels of *ADRB1* than those in the immature hPSC-derived cells (Supplementary Fig. 9a) indicating that they should be able to respond to appropriate stimuli.

To induce a pathological response in the mature cardiomyocytes we cultured them in the presence of isoproterenol (100 uM), a small molecule adrenergic agonist in a hypoxic (5% O2) environment, in media containing low glucose (2.0 g/L) and Palmitate (200 uM) for 6 days (Supplementary Fig. 9b). Immature control cells were treated under the same conditions. This treatment induced the upregulation of expression of the glycolysis-related genes *GLUT1, GAPDH*, and *LDHA* in the mature cardiomyocytes suggesting that they are undergoing a switch in metabolism (Fig. 8a). The immature cells showed similar changes in *GLUT1* and *LDHA* expression (Fig. 8a). To assess the glycolytic flux in these cells, we measured the extracellular acidification rate (ECAR) as an index of glycolytic activity using the Seahorse XF assay. Mature cardiomyocytes exposed to the pathological stimuli showed a rapid increase in

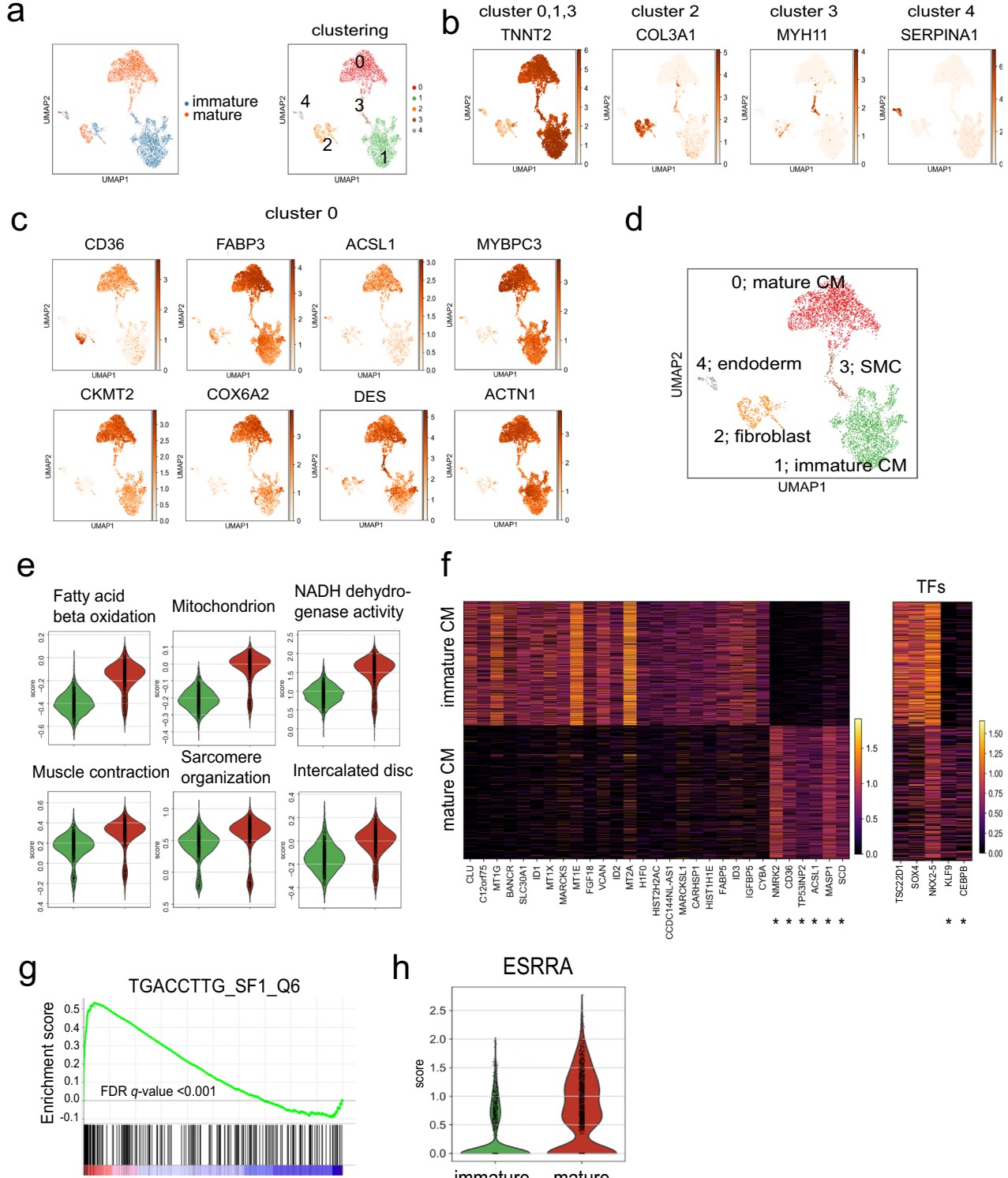

**Fig. 5 Single cell RNA sequencing analyses of mature compact cardiomyocytes. a** UMAP plot of the combined data set of day 32 immature and mature cardiomyocytes showing 5 different clusters. UMAP plots of the expression patterns of representative genes in the different clusters (**b**) and of the expression patterns of representative genes found at higher levels in cluster 0 than the other clusters (**c**). **d** UMAP plot indicating the lineage composition of the 5 different populations. **e** Violin plots of the Gene Ontology (GO) analyses comparing the mature (cluster 0, red) to the immature cardiomyocytes (cluster 1, green). **f** Heatmap visualization of marker genes (left) and transcription factors (right) using a binary enrichment search. *maturation signature genes. **g** Enrichment score plot of the TGACCTTG_SF1_Q6 gene set. **h** Violin plot of *ESRRA* expression in the immature (cluster 1) and mature (cluster 0) cardiomyocytes.

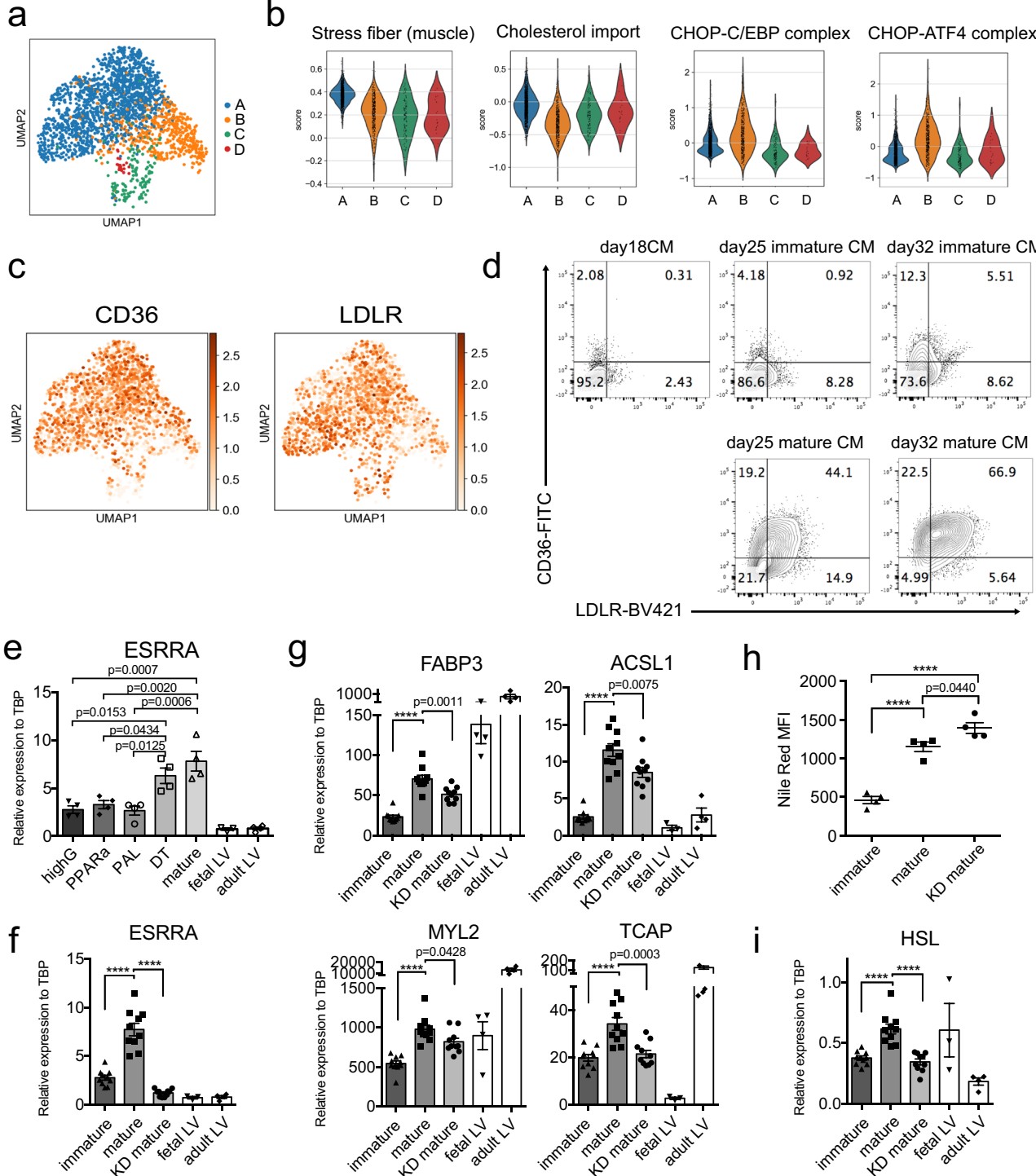

**Fig. 6 Detailed molecular analyses of the mature cardiomyocyte population. a** UMAP plot of mature cardiomyocytes showing 4 different clusters. **b** Violin plots of the Gene Ontology (GO) analysis comparing expression patterns in the indicated clusters. **c** UMAP plot showing expression patterns of CD36 and LDLR. **d** Representative flow cytometric analyses of CD36 and LDLR expression in the immature and mature cardiomyocytes on the indicated days of differentiation. **e** RT-qPCR expression analyses of *ESRRA* in day 32 cardiomyocytes cultured in the indicated conditions (*N* = 4). **f** RT-qPCR expression analyses of *ESRRA* in the immature, mature, and mature ESRRA knock down cardiomyocytes (KD mature) (*N* = 10). **g** RT-qPCR expression analyses of FAO (*FABP3, ACSL1*) and sarcomere (*MYL2, TCAP*)-related genes in the immature, mature, and KD mature cardiomyocytes (*N* = 10). **h** Flow cytometric analyses of Nile Red staining in the indicated cell populations (*N* = 4). **i** RT-qPCR expression analyses of HSL in the immature, mature, and KD mature cardiomyocytes (*N* = 10). All statistical analyses were performed by one-way ANOVA with Tukey's multiple comparisons (****$p$ < 0.0001). Data are presented as mean values ± SEM. Fetal LV and adult LV tissues were included as a reference for in vivo expression in the RT-qPCR analyses. Fetal LV: Fetal left ventricular tissue (*N* = 3−4), Adult LV: Adult left ventricular tissue (*N* = 4).

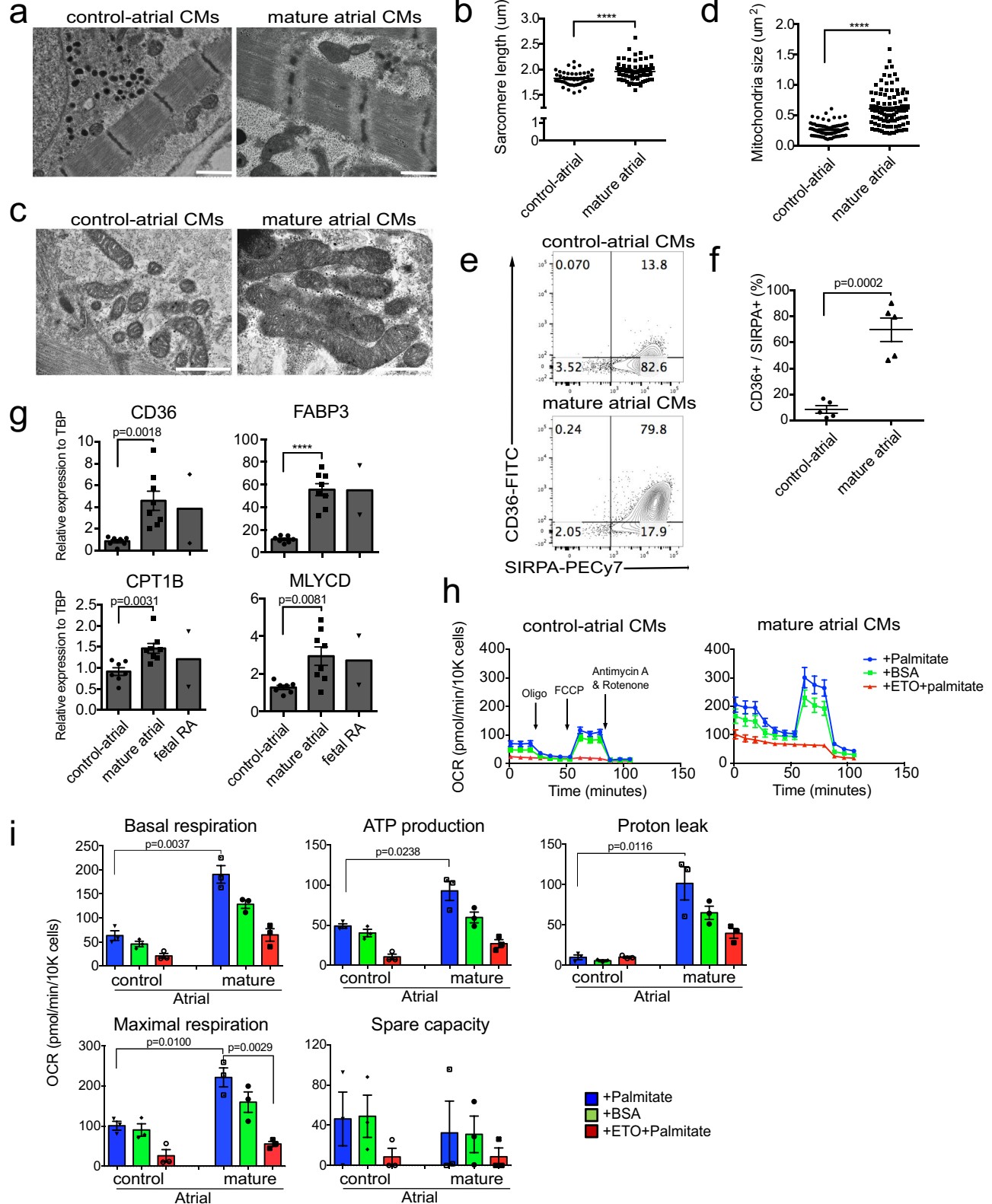

ECAR after the injection of glucose, and a rapid decrease in ECAR following the injection of 2-deoxy-glucose (2-DG), an inhibitor of glycolysis (Fig. 8b). This response was not detected in the untreated mature cardiomyocytes indicating that glycolytic activity was significantly upregulated by the pathological stimuli. Exposure to the pathological stimuli also enhanced glycolysis in the immature cells (Fig. 8b), which

might be expected as these cells are dependent on glucose metabolism.

In addition to changes in metabolism, we also detected increases in lipid accumulation in the treated populations as demonstrated by Nile red staining (Fig. 8c). Expression of genes associated with triacylglycerol synthesis, *CD36* and *GPD1* as well as Perilipin 2 (*PLIN2*) that plays a role in abnormal lipid

**Fig. 7 Maturation of atrial cardiomyocytes. a** Representative TEM images of sarcomere structures in the control (immature) and PPDT/PAL treated (mature) atrial cardiomyocytes. Scale bar; 1 um. **b** Quantification of sarcomere length (based on TEM analyses) in immature and mature atrial cardiomyocytes ($N = 49$ cells from control atrial and 74 cells from mature atrial examined over 3 independent experiments). **c** Representative TEM images of mitochondria structure in control and mature atrial cardiomyocytes. Scale bar; 1 um. **d** Quantification of mitochondria size in control and mature atrial cardiomyocytes ($N = 79$ cells from control atrial and 105 cells from mature atrial examined over 3 independent experiments). **e** Representative flow cytometric analyses of CD36 and SIRPA expression in control and mature atrial cardiomyocytes. **f** Quantification of the proportion of CD36+/SIRPA+ cells in control and mature atrial populations ($N = 5$). **g** RT-qPCR expression analyses of FAO-related genes in control and mature atrial cardiomyocytes ($N = 7$ biologically independent experiments from control atrial and $N = 8$ from mature atrial). Fetal RA tissues were included as a reference for in vivo expression. **h** Representative kinetics of the oxygen consumption rate (OCR) measured with the FAO Cell Mito stress test assay in control and mature atrial cardiomyocytes (control-atrial; data from 10 technical replicates in palmitate, 10 technical replicates in BSA, 2 technical replicates in ETO, mature atrial; data from 9 technical replicates in palmitate, 10 technical replicates in BSA, 2 technical replicates in ETO). **i** Comparison of each parameter of the FAO Cell Mito stress assay in control and mature atrial cardiomyocytes ($N = 3$ biologically independent experiments). OCR in each parameter measured with palmitate (blue) was compared by two-sided unpaired $t$-test and maximal respiration in mature atrial cells was compared by one-way ANOVA with Tukey's multiple comparisons. All other statistical analyses (**b**, **d**, **f**, **g**) were performed by two-sided unpaired $t$-test (****$p < 0.0001$). Data are presented as mean values ± SEM. Fetal RA: Fetal right atrial tissue ($N = 2$).

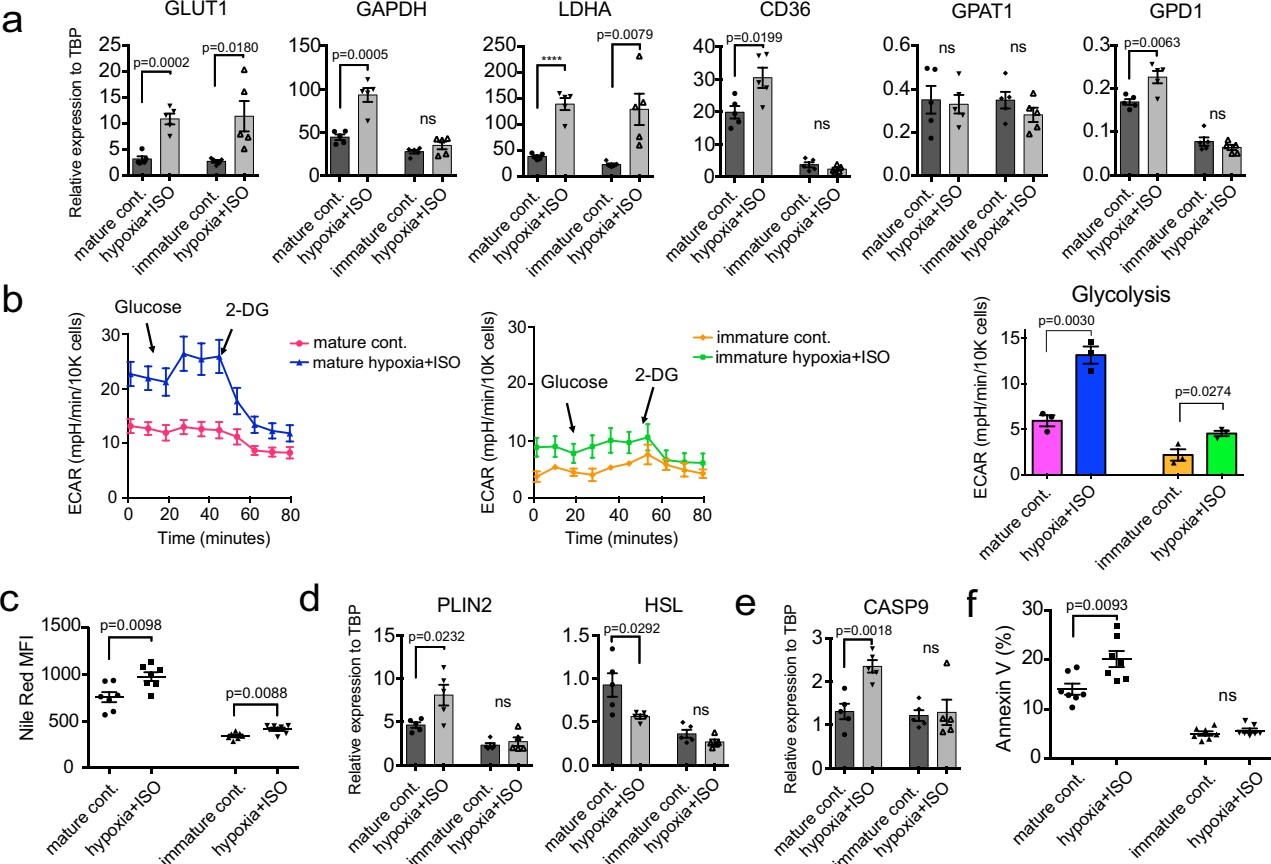

**Fig. 8 Modeling pathological adaptation using mature compact cardiomyocytes. a** RT-qPCR expression analyses of glycolysis-related and TAG synthesis-related genes in untreated and hypoxia+ISO-treated immature and mature ventricular cardiomyocytes ($N = 5$). **b** Left; Representative kinetics of ECAR measured by the seahorse XF assay in untreated and hypoxia+ISO-treated immature and mature ventricular cardiomyocytes (mature control; data from 3 technical replicates, mature hypoxia+ISO; data from 3 technical replicates, immature control; data from 3 technical replicates, immature hypoxia +ISO; data from 3 technical replicates). Right; Quantification of glycolysis based on ECAR measurement with the seahorse XF assay in the indicated cardiomyocyte populations ($N = 3$ biologically independent experiments). **c** Flow cytometric analyses of Nile Red staining in the indicated populations ($N = 7$). **d** RT-qPCR expression analyses of *PLIN2* and *HSL* and **e** *CASP9* in the indicated populations ($N = 5$). **f** Flow cytometric-based quantification of the proportion of Annexin V+ cells in untreated and hypoxia+ISO-treated immature and mature ventricular cardiomyocyte populations ($N = 7$). All statistical analyses were performed by two-sided unpaired $t$-test (****$p < 0.0001$). Data are presented as mean values ± SEM. ns not significant.

accumulation[49] were only upregulated in the mature cardiomyocytes cultured under the pathological stimuli (Fig. 8a, d). Correspondingly, expression of *HSL* was downregulated only in the treated mature cardiomyocytes (Fig. 8d). Analyses of expression of the apoptosis-related gene *CASP9* and the

proportion of Annexin V+ cells revealed that isoproterenol/ hypoxia treatment induced apoptosis in the mature but not in the immature population (Fig. 8e, f, Supplementary Fig. 9c). Taken together, these findings show that it is possible to model pathological responses in the mature cardiomyocytes and that

these cells recapitulate changes associated with heart failure including activation of glycolysis, lipid accumulation and apoptosis, as summarized in Supplementary Fig. 9d.

**Engraftment of mature and immature cardiomyocytes.** To determine if the maturation status of the cells can influence their ability to engraft heart tissue in vivo, we transplanted both the mature and immature populations into infarcted nude rat hearts and analyzed the graft quality by immunohistology at 2 and 8 weeks post-transplantation. Freshly prepared mature and

immature cells were used for these studies as the mature cardiomyocytes did not survive the cryopreservation process well. Grafts containing cTNT+ cardiomyocytes were detected in most of the transplanted rats at both time points (5 out of 6 rats at 2 weeks, 8 out of 9 rats at 8 weeks in mature cell transplantation group and 8 out of 8 rats at 2 weeks, 8 out of 8 rats at 8 weeks in immature cell transplantation group, respectively) (Fig. 9a, Supplementary Fig. 10a). At 2 weeks post transplantation, the size of the grafts generated by the two cardiomyocyte populations was comparable (Fig. 9a, b). Detailed analyses showed that the cardiomyocytes in the grafts from the mature cells had significantly

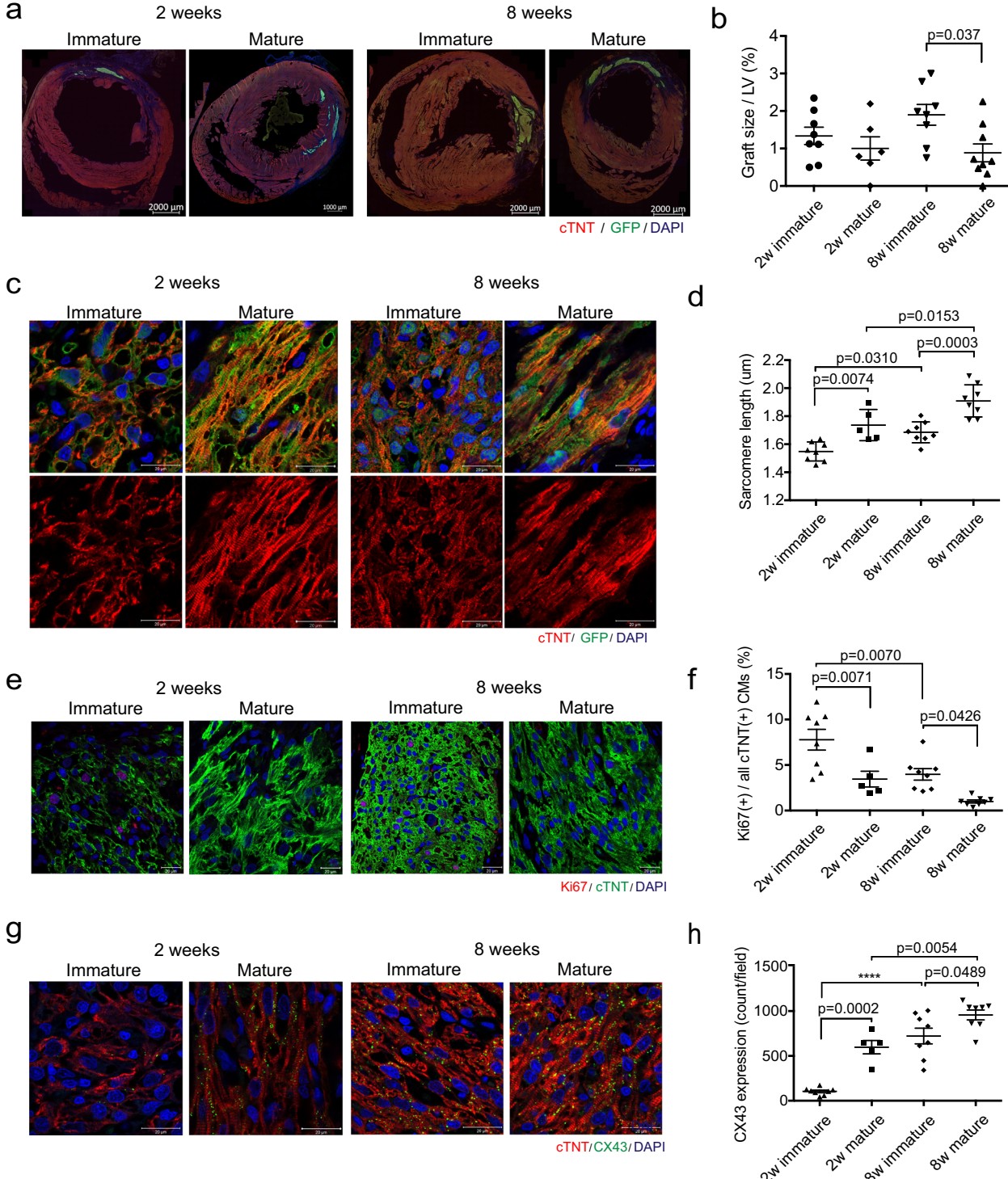

**Fig. 9 Engraftment of mature and immature cardiomyocytes into infarcted rat hearts. a** Representative immunostaining of whole rat hearts 2 and 8 weeks following transplantation of the mature and immature cardiomyocyte populations. **b** Quantification of graft size generated from the transplanted mature and immature cells ($N = 8$ recipients of 2-week immature cells, 6 recipients of 2-week mature cells, 8 recipients of 8-week immature cells, and 9 recipients of 8-week mature cells). **c** Representative immunostaining of human cardiomyocyte grafts 2 and 8 weeks following transplantation of the indicated cardiomyocyte populations. Scale bar: 20 um. **d** Quantification of sarcomere length in cardiomyocytes in grafts (2 and 8 weeks) generated from immature and mature cells ($N = 8$ recipients of 2-week immature cells, 5 recipients of 2-week mature cells, 8 recipients of 8-week immature cells, and 8 recipients of 8-week mature cells). **e** Representative immunostaining showing Ki67+ cells in cardiomyocyte grafts generated from immature and mature cells. Scale bar: 20 um. **f** Quantification of the percentage of Ki67+ cardiomyocytes in grafts (2 and 8 weeks) derived from immature and mature cells ($N = 8$ recipients of 2-week immature cells, 5 recipients of 2-week mature cells, 8 recipients of 8-week immature cells, and 8 recipients of 8-week mature cells). **g** Representative immunostaining showing CX43 expression in grafts derived from immature and mature cells. Scale bar: 20 um. **h** Quantification of CX43 expression in grafts (2 and 8 weeks) derived from immature and mature cells ($N = 8$ recipients of 2-week immature cells, 5 recipients of 2-week mature cells, 8 recipients of 8-week immature cells, and 8 recipients of 8-week mature cells). Quantification of the sarcomere length, the proportion of Ki67+ cardiomyocytes, and CX43 expression was evaluated from 5 to 10 randomly selected areas of cTNT+ grafted cardiomyocytes and averaged for each transplanted heart. All statistical analyses were performed by one-way ANOVA with Tukey's multiple comparisons (****$p < 0.0001$). Data are presented as mean values ± SEM.

longer sarcomeres than those in the immature grafts (Fig. 9c, d). Additionally, the grafts from the mature cells contained fewer proliferating cells as measured by Ki67 and phosphorylated histone H3 (pH3; a mitotic marker) and more connexin 43 (CX43) protein than those derived from the immature cells (Fig. 9e–h, Supplementary Fig. 10b, c). Differences in CX43 were already detected in the populations prior to transplantation as the day 32 mature embryoid bodies (EBs) showed higher levels of CX43 message and contained more CX43 protein than immature EBs (Supplementary Fig. 10d, e). The presence of CX43 is important as it is essential for the formation of gap junctions and electrical integration in the heart[50]. One of the commonly used measures of cardiomyocyte maturation is the switch from the fetal to the adult isoform of the sarcomeric protein TNNI[42]. Analyses of TNNI expression showed that the cells in the 2-week grafts from both cell populations expressed only the fetal isoform (TNNI1) indicating that this aspect of cardiac development was not fully mature at this time point (Supplementary Fig. 10f). At 8 weeks following transplantation, the grafts from the immature cells were modestly larger than those generated from the mature populations (Fig. 9b). The cells in both the mature and immature grafts had further matured between 2 and 8 weeks as demonstrated by changes in most of the above parameters (Fig. 9a–h). Although the changes were most pronounced in the immature cell-derived grafts, the cells in the mature grafts still showed a more mature profile at the 8-week time point as demonstrated by the presence of longer sarcomeres, a lower proliferative index and higher levels of CX43 than those in the immature grafts (Fig. 9a–h, Supplementary Fig. 10b, c). Despite the increased levels of CX43 protein, it remained distributed throughout the graft and was not aligned at the cell-cell junctions as observed in the host rat myocardium indicating that human tissue was not yet fully mature (Supplementary Fig. 10g). The 8-week immature grafts contained a higher percentage of Ki67 and pH3-positive cardiomyocytes than the mature grafts, an observation that could account for their larger size. Cells in both the mature and immature 8-week grafts upregulated expression of TNNI3 the adult isoform of TNNI. However, both populations also continued to express the fetal isoform (TNNI1) demonstrating that this aspect of maturation was not yet complete (Supplementary Fig. 10f).

Collectively, these findings show that the mature cardiomyocytes generated more mature grafts than the immature cardiomyocytes, providing evidence that the maturation status of the transplanted cells can impact the quality of the graft.

## Discussion

In this study, we used insights from our understanding of cardiomyocyte development and maturation in the fetus and neonate to design a staged protocol to generate mature compact cardiomyocytes from hPSCs. Using flow cytometric analysis of CD36 expression as a primary screen for maturation, we found that the combination of a PPARa agonist, dexamethasone, T3 and palmitate in media containing low glucose was most effective at inducing a mature phenotype in the hPSC-derived ventricular cardiomyocytes. These regulators were selected, as the levels of PPAR signaling as well as the concentrations of T3 and corticosteroids increase dramatically in different model organisms and in humans during the perinatal period. Additionally, genetic studies have shown that signaling through the PPAR pathway as well as responsiveness to hormones and lipids regulate cardiomyocyte maturation in vivo[12–16,38,39]. A number of previous studies have addressed the question of metabolic maturation of hPSC-derived cardiomyocytes and have shown that treatment with different combinations of the above stimuli did promote some degree of maturation in vitro[19–22,33]. While changes were well documented, comparisons of the effect of the manipulations and the degree of maturation between studies is difficult as quantitative analyses were not used to evaluate entire populations. Through our use of flow cytometric analyses, we were able to compare different combinations of stimuli and show that PPDT was significantly more efficient than DT which in turn was more efficient than Pal alone at generating CD36+ cells. Beyond upregulation of CD36, the PPDT-treated cells expressed higher levels of key metabolic genes, were larger, had longer sarcomeres, greater mitochondrial mass and showed better contraction force than those treated with either DT or Pal alone. A summary of the differences observed in the cardiomyocytes induced under these different conditions is shown in Table 1. The correlation of CD36 expression with other parameters of maturation validates the use of this approach for quantifying maturation in populations of cardiomyocytes. This notion is supported by findings from a recent study which showed that CD36+ cardiomyocytes isolated from long-term cultured hPSC-derived cardiovascular populations displayed more mature properties than the CD36− cells[51].

For maturation protocols to be most useful, they need to be evaluated and optimized on defined cardiomyocyte populations. Previous studies with hPSCs have not used well characterized populations and as a consequence it is not known if the protocols described to date promote maturation in different subsets of cardiomyocytes. Here, we developed and optimized our PPDT/PAL protocol using compact ventricular cardiomyocytes that were specified through activation of the Wnt and IGF pathways known to be required for proliferation and expansion of the compact lineage in vivo[5,7]. When tested on atrial cardiomyocytes, PPDT/PAL also promoted their maturation, albeit to a lesser extent than observed in the ventricular cells. The smaller changes

**Table 1 A summary of the differences observed in the cardiomyocytes induced with PAL, DT, and PPDT/PAL compared to untreated control cells.**

|  | PAL | DT | PPDT/ PAL |
|---|---|---|---|
| FFA transporter (CD36) | No change | ++ | +++ |
| FFA mitochondria transporter (CPT1B) | No change | − | +++ |
| FAO intracellular regulator (MLYCD) | No change | No change | +++ |
| Endogenous FA utilization | + | +++ | +++ |
| Exogenous FA utilization | + | − | +++ |
| Intracellular lipid storage | No change | +++ | +++ |
| Hypertrophic change | + | ++ | +++ |
| Sarcomere organization | + | + | +++ |
| Mitochondria organization | + | ++ | +++ |
| $Ca^{2+}$ handling | No change | +++ | +++ |
| Contraction force | No change | No change | +++ |

*No change* not significantly changed compared to control (high Glucose media), + significantly higher compared to control (high Glucose media), − significantly lower compared to control (high Glucose media).

observed in the atrial cells may reflect the fact that atrial cardiomyocytes display lower metabolic activity than ventricular cardiomyocytes[52].

Our Seahorse analyses showed that although the cells treated with either PPDT or DT for 2 weeks were capable of undergoing FAO, they were not dependent on exogenous FAs, indicating they were using an endogenous source of lipids. The distinction between exogenous and endogenous lipid use is important as the ability to oxidize exogenously supplied lipids can be considered as the final step in metabolic maturation. The endogenous lipid source in the cells treated with either PPDT or DT in our study is likely the lipid droplets. Similar droplets have been detected in neonatal cardiomyocytes suggesting that cells in the heart rely on an endogenous source of lipids as they undergo the transition from glycolysis to FAO[10]. Our findings strongly suggest that the hyperactive stressed phenotype observed in the 14-day PPDT-treated cells is likely due to prolonged hormonal exposure as restriction of the duration of PPARa/T3/Dex treatment led to the development of a population that displayed spare capacity and the ability to oxidize exogenous FAs. This restriction in the timing of treatment more accurately recapitulates the transient perinatal spikes in the levels of T3 and corticosteroids observed in vivo during the first several weeks of life. Increases in FAO, mitochondrial biogenesis and ROS collectively have been shown to inhibit cardiomyocyte proliferation[15,36]. Similar inhibition may be occurring in vitro, as the PPDT/PAL treated population contained very few proliferating cells and was refractory to the proliferative stimuli provided by CHIR and IGF2. Combined with their increase in size, these characteristics suggest that these cells are transitioning from the hyperplasia stage (fetal) of cardiovascular development when increases in heart size result from proliferation to the hypertrophy stage (postnatal) when increases in size are due to increases in the size of the cardiomyocytes.

The findings from our scRNAseq analyses showed that the majority of cells in the population upregulated genes indicative of metabolic and structural maturation supporting the utility of CD36 analysis as a method to track population-wide responses to the stimuli. The demonstration that the mature cells upregulate LDLR and that its expression can be measured by flow cytometry provides a second surface marker for monitoring cardiomyocyte maturation in the differentiation cultures. Although broad changes in maturation were observed in the PPDT/PAL-treated cells they failed to undergo the switch in expression of the TNNI

isoforms in vitro and only upregulated expression of the adult form following transplantation in vivo. One interpretation of these findings is that the upregulation of TNNI3 expression requires mechanical force and/or other factors that are not be provided in the EBs in vitro. The observation that the mature cells show increased expression of ESRRA is consistent with previous studies that have shown an upregulation of TGACCTTG/ESRRA during the fetal to postnatal transition[53] and with the recent findings of Sakamoto et al. that demonstrated that this receptor, together ESRRG is required for the maturation of mouse cardiomyocytes in vivo[46]. Although the function of these receptors is not completely understood, evidence suggests that ESSRA interacts with PGC1a/PPARA to induce global metabolic changes and structural maturation[54]. Our KD studies showed that ESRRA plays some role in maturation of the human cardiomyocytes but also indicated that other pathways are involved in this process. The finding that PPARA signaling failed to induce ESRRA expression and that the levels of PPARA were not reduced in the KD cells, suggests that under the conditions used in our study, these receptors are not regulating each other, but are likely functioning in parallel. Our global analyses of HES2-derived cells identify a set of genes that can be used as a molecular signature of metabolically mature ventricular cardiomyocytes. The demonstration that the ESI-17-derived mature cardiomyocytes upregulated the genes in this signature verifies their utility for monitoring the maturation status of cardiomyocytes produced from different hPSC lines as well as for comparing populations generated with different protocols in different labs.

One of the goals of generating mature cardiomyocyte populations is to establish platforms to model and treat cardiovascular diseases with target cell populations that approximate those found in the adult heart. To date, most disease modeling studies have used immature cells that may not accurately recapitulate the complex disease processes observed in the adult[55]. The observation that the PPDT/PAL matured cells progressed to a 'disease phenotype' in response to culture conditions that mimic the environment of the failing heart indicates that the maturation protocol developed here gives rise to cardiomyocytes that can be used to study cardiac pathologies beyond those resulting from genetic disorders. The ability to induce maturation in engineered biowire tissues demonstrates that this approach can also be used to model disease in 3D cardiovascular tissue; a format that enables the study of the role of different cell types in the disease process. Transplantation of hPSC-derived cardiomyocytes represents a promising cell therapy to remuscularize the human ventricle following MI. Studies with large animals have shown that following transplantation, immature cardiomyocytes will engraft the infarct induced scar and generate myocardium that integrates with the host tissue[56,57]. However, in all cases, the animals showed ventricular arrhythmias for the first few weeks following transplantation, possibly due to the immature nature of the transplanted cells. The finding that the metabolically mature cardiomyocytes generate grafts that contain higher numbers of connexin 43[+] cells, fewer proliferating cells and more structurally mature cells than those from the immature cardiomyocytes suggests that manipulation of the population prior to transplantation may be one approach to mitigate the arrhythmias and develop a safer therapy. As the protocol in this study was designed for maturation of cardiomyocytes in EBs in suspension culture, it is easily scalable for the production of the large number of cells required for transplantation.

## Methods

**Directed differentiation of hPSCs.** For ventricular differentiation, we used a modified version of our embryoid body (EB)-based protocol[58]. hPSC populations (HES2, ESI-17) were dissociated into single cells (TrypLE, ThermoFisher) and

re-aggregated to form EBs in StemPro-34 media (ThermoFisher) containing penicillin/streptomycin (1%, ThermoFisher), L-glutamine (2 mM, ThermoFisher), transferrin (150 mg/ml, ROCHE), ascorbic acid (50 mg/ml, Sigma), and mono-thioglycerol (50 mg/ml, Sigma), ROCK inhibitor Y-27632 (10 uM, TOCRIS) and rhBMP4 (1 ng/mL, R&D) for 24 h on an orbital shaker (70 rpm). At day 1, the EBs were transferred to mesoderm induction media consisting of StemPro-34 with the above supplements (-ROCK inhibitor Y-27632) and rhBMP4 (8 ng/ml), rhActivinA (12 ng/ml, R&D) and rhbFGF (5 ng/ml, R&D). At day 3, the EBs were harvested, dissociated into single cells (TrypLE), and re-aggregated in cardiac mesoderm specification media consisting of StemPro-34, the Wnt inhibitor IWP2 (1 uM, TOCRIS) and rhVEGF (10 ng/ml, R&D). At day 5, the EBs were transferred to StemPro-34 with rhVEGF (5 ng/ml) for another 5 days and then to DMEM high glucose (4.5 g/l, ThermoFisher) media with compact factors (CHIR (1 uM, TOCRIS), IGF2 (25 ng/ml, R&D)) and human insulin (10 ng/ml, Sigma) at day 10 for another 6 days. From day 16 to day 18, the EBs were transferred to DMEM high glucose media with XAV (4 uM, TOCRIS) and then transferred to maturation media [DMEM containing low glucose (2 g/L) with Palmitic acid (200 uM, Sigma), Dexamethazone (100 mg/ml, Bioshop), T3 hormone (4 nM, Sigma) and GW7647 (PPARA agonist, 1 uM, Sigma] for the following 9 days. Finally, the EBs were cultured in DMEM containing low glucose with Palmitate (200 uM) alone for the following 5 days (total 32 days). Cultures were incubated in a low oxygen environment (5% $CO_2$, 5% $O_2$, 90% $N_2$) for first 10 days and a normoxic environment (5% $CO_2$, 20% $O_2$) for the following 22 days. From day 10 to day 32, the EBs were cultured in polyheme coated low binding 10 cm culture dishes on an orbital shaker (70 rpm). For atrial differentiation, we used different concentrations of rhBMP4 (3 ng/ml) and rhActivinA (1 ng/ml) from day 1 to day 3, followed by Retinoic Acid (0.5 uM, Sigma) from day 3 to day 5 as previously described[17]. For the atrial maturation process, we used the same maturation media as for ventricular maturation from day 18 to day 32.

**Flow cytometry.** The EBs were dissociated by incubation in Collagenase type 2 (0.5 mg/ml, Worthington) in HANKs buffer overnight at room temperature followed by TrypLE for 5 min at 37 °C. The following antibodies were used for staining: anti-SIRPa-PeCy7 (Biolegend, 1:1000), anti-CD36-FITC (BD PharMingen, 1:200), anti-LDLR-BV421 (BD PharMingen, 1:100), anti-cardiac isoform of cTNT (ThermoFisher Scientific, 1:2000), or anti-myosin light chain 2 (Abcam,1:1000), anti-CD90-APC (BD PharMingen, 1:1000). For unconjugated primary antibodies, the following secondary antibodies were used for detection: goat anti-mouse IgG-APC (BD PharMingen, 1:500), or donkey anti-rabbit IgG-PE (Jackson ImmunoResearch, 1:500). For cell-surface marker analyses, cells were stained for 30 min at 4 °C in FACS buffer consisting of PBS with 5% fetal calf serum (FCS) (Wisent) and 0.02% sodium azide. For intracellular staining, cells were fixed for 20 min at 4 °C with 4% PFA in PBS followed by permeabilization using 90% methanol for 20 min at 4 °C. Cells were washed with PBS containing 5% FCS and stained with unconjugated primary antibodies in FACS buffer overnight at 4 °C. Stained cells were washed with PBS with 5% FCS and stained with secondary antibodies in FACS buffer for 30 min at 4 °C. Stained cells were analyzed using the LSR II Flow cytometer (BD PharMingen). Data were analyzed using FACS DIVA software (BD) and FlowJo software (Tree Star). Gating strategy is shown in Supplementary Fig. 11a, d, f, g.

**Immunohistochemistry.** The EBs were dissociated as described above and the cells were plated onto 24 well culture dishes pre-coated with matrigel (25% v/v, BD PharMingen). Cells were cultured for 2–3 days and were fixed with 4% PFA in PBS for 15 min at room temperature. Cells were permeabilized and blocked with PBS containing 5% donkey serum, 0.1% TritonX. The following antibodies were used for staining: mouse anti-cardiac isoform of cTNT (ThermoFisher Scientific, 1:200), rabbit anti-human HEY2 (Proteintech, 1:100), mouse anti-human ANF (Abcam, 1:100), rabbit anti-human cTNT (abcam, 1:200), or rabbit anti-human CD90 (abcam, 1:200). For detecting unconjugated primary antibodies, the following secondary antibodies were used: donkey anti-mouse IgG-Alexa488 (ThermoFisher, 1:500), donkey anti-rabbit IgG-Alexa555 (ThermoFisher, 1:500), donkey anti-rabbit IgG-Alexa488 (ThermoFisher, 1:500), donkey anti-mouse IgG-Alexa555 (ThermoFisher, 1:500). Cells were stained with primary antibodies in staining buffer consisting of PBS with 0.1% TritonX, and 5% donkey serum overnight at 4 °C. The stained cells were washed with PBS. The cells were then stained with secondary antibodies in PBS containing 0.1% BSA for 1 h at room temperature followed by DAPI staining. For paraffin sections, tissues were fixed by 4% PFA and embedded. After the deparaffinization and rehydration, heat-induced epitope retrieval was performed followed by immunostaining. Stained cells were analyzed using an EVOS Microscope (ThermoFisher) and Zeiss LSM700 confocal microscope (Zeiss).

**Quantitative real-time PCR.** Total RNA from samples was isolated using RNAqueous-micro Kit including RNase-free DNase treatment (Invitrogen). RNA from dissected ventricular and atrial tissue of human fetal hearts (gestation week 17) was isolated using the TRIzol method (ThermoFisher) and treated with DNase (Ambion). The work with human fetal tissue was approved by the Research Ethics Board of the University Health Network, Toronto. Isolated RNA was reverse

transcribed into cDNA using oligo (dT) primers and random hexamers and iscript Reverse Transcriptase (ThermoFisher). qRT-PCR was performed on an EP Real-Plex MasterCycler (Eppendorf) using a QuantiFast SYBR Green PCR kit (QIA-GEN). The copy number of each gene relative to the house keeping gene *TBP* is shown. Primer sequences are listed in Supplementary Table 5. Fetal heart tissues were included as a reference for in vivo expression. Informed consent was obtained from the participants.

**Transmission Electron Microscope (TEM).** The samples were fixed in 2.5% glutaraldehyde in PBS, rinsed and post-fixed in 1% $OsO_4$ (Electron Microscopy Sciences) for 1 h. The tissue was again rinsed with 0.1 M Sorenson's Phosphate buffer, dehydrated through an ascending ethanol series, then infiltrated with and embedded in modified Spurr's resin. From the area of interest, identified by thick sectioning, ultrathin sections (90–100 nm) were cut with a Leica UC6 ultra-mictrotome (Leica). Thin sections were stained with Uranyless and lead citrate and then examined using a Hitachi HT7700 transmission electron microscope (Hitachi). Analysis were performed from 3 to 5 independent experiments.

**Seahorse OCR / ECAR measurement.** For the seahorse XF FAO assay, a few EBs were plated onto an XFe24 cell culture microplate coated by Matrigel 48 h prior to the assay. 24 h prior to the assay, we replaced the culture media with substrate-limited medium containing 0.5 mM glucose (Sigma), 1.0 mM Glutamine (Life technology), 0.5 mM Carnitine (Sigma), and 1% Fetal Bovine Serum (Wisent) in DMEM no glucose media (ThermoFisher). 45 min prior to the assay, we washed the cells two times with FAO assay medium, added 375 μl/well FAO assay medium to the cells and incubated for 45 min at 37 °C. We loaded the assay cartridge with XF Cell Mito Stress Test compounds (2 μM oligomycin, 5 μM FCCP, 0.5 μM rotenone/0.5 μM antimycin A). 15 min prior to starting the assay, we added 37.5 μl etomoxir (Sigma, 40 μM) or vehicle to each well and then incubated them for 15 min at 37 °C in a non-CO2 incubator. Just prior to starting the assay, we added 87.5 μl XF Palmitate-BSA FAO Substrate or BSA to the appropriate wells. The XF cell Culture Microplate was immediately inserted into the Seahorse XFe Analyzer and the XF Cell Mito Stress Test was run. After the measurement of OCR, EBs were dissociated and the cell number in each well was counted. OCR was normalized per 10,000 cells. For the glycolysis assay, a few EBs were similarly prepared onto a XFe24 cell culture microplate. Glucose (Sigma) and 2-DG (Sigma) were prepared and loaded into the assay cartridge (final concentration; glucose 10 mM, 2-DG 100 mM). Culture media was changed to glycolysis assay media and the XF cell Culture Microplate was inserted into the Seahorse XF24 Analyzer and the assay was run. After the measurement of ECAR, EBs were dissociated and the cell number in each well was counted. ECAR was also normalized per 10,000 cells. Data were analyzed using Wave software (Agilent).

**$Ca^{2+}$ transient measurement.** For the $Ca^{2+}$ transient measurement, the EBs were dissociated into single cell at day 30 and replated at a concentration of $2 \times 10^6$ cells onto 3.5 cm culture dishes coated by Matrigel. We cultured the cells in the monolayer format for 3 days in the different media and then loaded them with Fluo-4 (Invitrogen, final concentration; 4 uM) for 30 min at 37 °C on the day of the measurement. Prior to analyses, the cells are washed with culture media and then incubated in the same media for additional 30 min. At this point the cells are transferred to Tyrode buffer at 37 °C and then analyzed. For imaging, a Zoom microscope body MVX10 with the objective MVPLAPO 0.63X (NA 0.15, WD 87 mm, FN 22, Olympus) for an overall FOV of 10 mm x 10 mm was used with the wavelength 490–535 nm for the excitation and 532–588 nm for bandpass-filter Fluo-4. Cells are stimulated (1 Hz, pulse duration 5 ms, voltage 10 V) via electrodes using a PowerLab system. $Ca^{2+}$ transients are then measured using MetaMorph software after selecting a ROI per monolayer and manually determining the start and end of each $Ca^{2+}$ depolarization/repolarization through time. Data were collected from 8 to 20 samples in each condition.

**scRNA sequencing and analysis.** The EBs were dissociated as described above and the cells were stained with DAPI. DAPI- live cells were then sorted using FACSAriaRITT (BD PharMingen) at the Sickkids/UHN flow cytometry facility. After the live cell sorting, scRNA sequencing was performed using the 10x Genomics platform, sequenced on Illumina NovaSeq 6000, and analyzed using GRCH38 (hg38) as follows.

Single-cell RNA sequencing of day 20 ventricular cardiomyocytes were first filtered to remove lowly expressed genes (defined as those found in less than 3 cells) and damaged cells with high mitochondrial genome transcript content (defined as 12 median absolute deviations above the median to account for the typically high mitochondrial content in cardiomyocytes). The data set was then normalized using the deconvolution method implemented in the scran R package[59], which pools cells with similar gene expression profiles and library sizes together to normalize. We then performed principal component analysis on normalized data to reduce the number of dimensions in the data. The number of principal components to use in clustering and t-SNE was determined to be 17 by plotting the standard deviations of the first 30 components in a scree plot and selecting the point after which standard deviations are similarly minimal and thus would not contribute significantly to resolving variances between cells in downstream analyses. All cells

were then iteratively clustered in Seurat 2.0[60] at increasing resolutions until the number of differentially expressed genes between two neighboring clusters reached 0. We then chose the optimal clustering resolution, defined as the point where the number of clusters was maximized while the number of differentially expressed (DE) genes between neighboring clusters remained larger than 0 (using scClustViz, accessed at https://www.ncbi.nlm.nih.gov/pmc/articles/PMC6456841/), and annotated all clusters by examining expression of known marker genes. The chosen clustering resolution for the data set was 0.6. Finally, all cells that do not express TNNT2 were eliminated in the process of creating a cardiomyocyte-only map. Downstream differential expression analysis between clusters was done using the FindMarker function in Seurat. Results were visualized using base graphics in R.

Pathway enrichment analysis: Gene Set Variation Analysis (GSVA)[61] was used to identify the signaling pathways that are differentially regulated in the HEY2-high versus HEY2-low populations. Cells belonging to clusters 1 and 2 (colored as pink and yellow, respectively, in Fig. 1b) were characterized as HEY2-high, while cells belonging to cluster 4 (green, upper left) were characterized as HEY2-low. GSVA was run on cells in these three clusters using the Bader lab gene sets for biological processes without electronic annotation (file Human_GO_AllPathways_no_GO_iea_July_01_2018_symbol.gmt, accessed at http://download.baderlab.org/EM_Genesets/July_01_2018/Human/symbol/ Human_GOBP_AllPathways_no_GO_iea_July_01_2018_symbol.gmt). P-values and false discovery rates (FDR) for enriched pathways were determined using a simple linear model as implemented in the limma R package[62]. Pathways with FDR less than 0.05 were determined as differentially enriched between HEY2-high and HEY2-low populations. GSVA results were then visualized using EnrichmentMap in Cytoscape[63].

For the analysis of day 32 immature and mature cells, we performed the analysis as follows.

Software Tools: Python (v3.7), Scanpy (v1.4.4)[64] and GOATOOLS (v0.9.7)[65], and their necessary dependencies, were used for these single cell RNA sequencing analyses.

Data Preprocessing: Raw data consisted of CellRanger-processed, filtered feature matrices. Datasets were read in and concatenated using scanpy. read_10x_mtx. The combined dataset was then filtered to only contain cells with a minimum of 1500 genes and to remove genes not present in at least 10 cells using scanpy.preprocessing.filter_cells and scanpy.preprocessing.filter_cells functions. The data set was then normalized and log(x + 1) transformed using the default settings in the scanpy.preprocessing.normalize_per_cell and scanpy.preprocessing. log1p functions. Principal component analysis was then performed on the transformed data using the 'arpack' solver with otherwise default settings using the scanpy.tools.pca function. Finally, a neighborhood graph was computed using the scanpy.preprocessing.neighbors function with 10 neighbors specified, otherwise default settings were used.

Dimensionality Reduction, Clustering and Differential Expression: UMAP coordinates were calculated using the scanpy.tools.umap function under default settings. Whole dataset clustering[66] was performed using scanpy.tools.leiden function with a resolution of 0.2 and otherwise default settings. The dataset was subsequently subset, selecting only cluster 0 from Fig. 5a. This subset data, representing mature cardiomyocytes, was then reprocessed in isolation. PCA analysis was performed again using default settings on the subset data using only highly variable genes detected using the scanpy.preprocessing. highly_variable_genes using default settings. A neighborhood graph was then computed using the scanpy.preprocessing.neighbors function with 5 neighbors and 25 principal components specified, otherwise default settings were used. Leiden clustering was performed on the reprocessed data with a resolution of 0.2. Differentially expressed genes were discovered using the scanpy.tl. rank_genes_groups function between the previously defined Leiden groups using default settings. All genes considered for further analysis had a positive absolute z-score (underlying the computation of the p-value for each gene for each group) of greater than 1.96.

Gene Ontology Analysis: Gene Ontology Enrichment Analysis (GOEA) was performed using GOATOOLS. GO ontologies and annotations were downloaded using the goatools.base.download_go_basic_obo and goatools.base. download_ncbi_associations functions respectively. Ontologies were then loaded in using the goatools.obo_parser.GODag function. Human GO associations were then selected and stored as a list of named tuples using the call 'taxids = [9606]' in the goatools.anno.genetogo_reader.Gene2GoReader function. Finally the background gene set, all human protein-coding genes, were loaded in using goatools.test_data. genes_NCBI_9606_ProteinCoding.GENEID2NT function. The human ontologies, associations, and background gene set were then used for GOEA. GOEA analysis was performed on the previously discovered differentially expressed gene sets using the goatools.goea.go_enrichment_ns.GOEnrichmentStudyNS.run_study function using default settings. Enriched ontologies with a Benjamini–Hochberg-corrected p-value less than 0.05 were retained for further analysis.

Enriched gene ontology gene lists were used to generate an enrichment score using the scanpy.tl.score_genes function with default settings. Violin plots were used to highlight the difference in gene set enrichment between the previously defined Leiden clusters.

Gene signature analysis and network analysis: GSEA (v4.0.3)[67] and Cytoscape (v3.8.0)[68] were downloaded and used as is for enrichment score and enrichment map visualization respectively.

Figure 5f: Gene signature analysis was performed using a binary enrichment search strategy. First, log-normalized data was binarized by setting all expression values greater than zero equal to one. Next, transcripts are filtered, retaining only transcripts that are present in at least 50% of the target batch and less than 25% in the remainder of the data. The binarized transcripts are summed across all cells in each batch and then normalized by dividing by the total number of cells considered in the summation. The binary enrichment ratio is then calculated by dividing the normalized sum of each group by that of the remainder of the data (all other groups). Finally, the binary enrichment ratio is calculated by dividing the normalized sum of the target batch by that of the remainder of the data. The first heatmap in Fig. 5f represents the results of this binary enrichment search wherein the transcripts were limited to those with a binary enrichment ratio of 3 or greater. The second heatmap was constructed using a similar scheme save the enrichment ratio was lower to 2 and the list was filtered to include only known transcription factors[69]. Both heatmaps were produced using scanpy.plotting.heatmap.

Figure 5g and Supplementary Fig. 5c: Normalized, log-transformed data was used for gene set enrichment analysis. The dataset was subset to only include mature and immature cardiomyocyte sub-clusters annotated in Fig. 5d. All genes in the subset data were then scored and ordered by their signal-to-noise ratio. Signal-to-noise was calculated between the subset clusters by dividing the mean difference of each gene by the sum of the standard deviations. The ordered gene list was imported into the GSEA software and a GSEAPreranked analysis performed using two gene set databases, the Hallmark gene set (h.all.v7.1.symbols) and the TFT_Legacy subset of TFT (c3.tft.tft_legacy.v7.1.symbols). The number of permutations used was 10,000 and all other settings were left in the default state. Enrichment score plots are automatically generated during the analysis and are presented as is.

Enrichment map was generated using the Cytoscape Enrichment Map Plugin in the GSEA software. For this analysis the similarity cutoff overlap coefficient was set to 0.1 and all other settings were left in the default state. Displayed is a subset of the resulting Enrichment Map annotated using the gene set names describing each node.

**Ki67 flow cytometry**. The EBs were dissociated as described above. The following antibodies were used for staining: mouse anti-Ki67 (DAKO, 1:100) and rabbit anti-cardiac isoform of cTNT (ThermoFisher Scientific, 1:500). The following secondary antibodies were used for detection: donkey anti-mouse IgG-APC (BD Pharmigen, 1:500), or donkey anti-rabbit IgG-PE (Jackson ImmunoResearch, 1:500). Cells were fixed for 20 min at 4 °C with 4% PFA in PBS followed by permeabilization with 90% methanol for 20 min at 4 °C. Cells were washed with PBS containing 5% FCS (Sigma) and stained with unconjugated primary antibodies in FACS buffer overnight at 4 °C. Stained cells were washed with PBS with 5% FCS and stained with secondary antibodies in FACS buffer for 30 min at 4 °C. Stained cells were analyzed using a LSR II Flow cytometer (BD PharMingen). Ki67+ (%) was measured within the cTNT+ populations. Data were analyzed using FlowJo software (Tree Star). The gating strategy is shown in Supplementary Fig. 11b.

**Nile red staining by flow cytometry**. To quantify the lipid droplets, Cayman's Lipid Droplets Fluorescence Assay Kit (Cayman) was used. After the EBs were dissociated as described above, cells were fixed with a Fixative Solution for 10 min at room temperature. The cells were then washed with the Assay Buffer and stained with the Nile Red Staining Solution at room temperature for 15 min. The cells were again washed with the Assay Buffer and analyzed with filter sets to detect FITC using a LSR II Flow cytometer (BD PharMingen). Mean Fluorescent Intensity (MFI) was measured using FlowJo software (Tree Star). The gating strategy is shown in Supplementary Fig. 11c.

**Contraction force measurement**. At day 18 of differentiation, the EBs were dissociated as described above. The biowire cardiac tissues were generated and analyzed as previously described[41]. In brief, cardiomyocytes and human cardiac fibroblasts (Lonza, #CC-2904) were mixed in a ratio of 4:1· and then seeded into the microwells in fibrin gels. 2 days after the generation of the tissues, the media was changed into the 4 conditions; the media was changed every 3 days thereafter. After 2 weeks of treatment, force measurements were performed. For the force assessment in the tissues, video recordings of the rod deflection were performed under electrical stimulation (1 Hz) using a Leica EC3 camera. Image analysis was performed using ImageJ. Tissue widths were measured at the middle of the tissue and at the PDMS rods. Rod deflection for passive force was measured as the distance between the PDMS rod in the tissue's relaxed state, and the PDMS rod at non-deflected position. For force of contraction (active force), peak rod deflection under electrical stimulation was measured. To measure active force, its total force (when a tissue is in an active contraction) minus the passive force (when it's relaxed) was calculated . The measurement of tissue widths and rod deflection measurements were performed by a person blinded to the conditions. Data were collected from 7 to 9 independent experiments. Force calculations were performed using the following formula: $f(x, y) = 1.55x + 0.00256x^2 + 0.002156xy$. The function f represents force (N), while x is the PDMS rod deflection from its non-deflected position origin, and y is the tissue width at the midspan of the PDMS rod. After the force measurement, tissues were fixed for immunohistochemistry or TEM analysis.

**siRNA-mediated knockdown of ESRRA**. siRNA transfection was carried out according to the manufacturer's instructions. Negative control (non-targeting), and ESRRA (TriFECTa DsiRNA Kit, IDT) siRNAs were transfected at 10 nM into dissociated day 18 cardiomyocytes using Lipofectamine RNAiMax (Invitrogen). The cells were then re-aggregated in low-binding 96 plate wells· for 2 days. Following this, the aggregates were collected and cultured in low-binding 10 cm dishes  and cultured according to the maturation protocol, shown in Fig. 3e, for 2 weeks. RT-qPCR and Nile red staining were then performed. To maximize the efficiency of the knockdown of ESRRA, three different specific siRNAs (supplied with the Kit) were transfected simultaneously.

**Annexin V apoptosis assay**. After the culture in the pathological conditions for 6 days, the EBs were dissociated as described above. To detect apoptosis following treatment with  the pathological stimuli, the TACS Annexin V assay (TREVIGEN) was performed by flowcytometry. Cells were washed with PBS and stained with Annexin V-FITC for 15 min at room temperature. Binding buffer was then added to samples prior to processing by flow cytometry. Stained cells were analyzed using the LSR II Flow cytometer (BD) and data were analyzed using FlowJo software (Tree Star). The gating strategy used is shown in Supplementary Fig. 11e.

**Cell transplantation into rat MI models**. All animal experimental protocols were approved by the Animal Use and Care Committee at the University Health Network. For the cell transplantation experiments, the HES2-GFP hESC line was used to generate the cardiomyocytes. HES2 hESCs were previously targeted at the human ROSA locus to enable constitutive eGFP expression[70]. The differentiated EBs were dissociated as described above and the cells directly transplanted without cryopreservation.

Rat myocardial infarction model: A permanent coronary ligation technique was used to generate a myocardial infarction in athymic nude rat hearts. All rats were intubated and positive pressure ventilation was maintained with a Harvard ventilator under anesthesia with inhalation of isoflurane (2–3%). The rat heart was exposed through a left anterolateral thoracotomy incision and a 7-0 suture was used to permanently ligate the left anterior descending artery.

Thoracotomy and cell transplantation: Cells were  transplanted  3–4 days after the induction of myocardial infarction. For the transplantation, rats were anesthetized and ventilated as described above. The heart was exposed and $10 \times 10^6$ cells (HES2-GFP differentiated cardiomyocytes) diluted by 75 µl Matrigel (100%, BD PharMingen) were injected using a 30 G needle into the infarcted region.

Sacrifice and analysis: 2 weeks or 8 weeks following the transplantation, the rats were sacrificed and their hearts were harvested. The hearts were fixed in 10% formaldehyde and embedded, and then sectioned horizontally at 6 levels to cover the entire LV area. After deparaffinization and rehydration, heat-induced epitope retrieval was performed followed by immunostaining. The following antibodies were used for immunostaining: mouse anti-cardiac isoform of cTNT (ThermoFisher Scientific, 1:200), rabbit anti-human cTNT (abcam, 1:200), rabbit anti-GFP (ROCKLAND, 1:200), mouse anti-Ki67 (DAKO, 1:100), rabbit anti-CX43 (abcam, 1:800), rabbit anti-pH3 (Cell Signaling, 1:200), rabbit anti-TNNI1 (NOVUS, 1:200), rabbit anti-TNNI3 (abcam, 1:200). For detecting unconjugated primary antibodies, the following secondary antibodies were used: donkey anti-mouse IgG-Alexa488 (ThermoFisher, 1:500), donkey anti-rabbit IgG-Alexa488 (ThermoFisher, 1:500), donkey anti-mouse IgG-Alexa555 (ThermoFisher, 1:500), or donkey anti-rabbit IgG-Alexa555 (ThermoFisher, 1:500). Sarcomere length was measured in cTNT$^+$ grafted cardiomyocytes randomly selected from 5 to 10 areas and averaged for each transplanted heart. CX43 expression was measured by counting the number of CX43$^+$ staining in one field of view (×40 magnification) of cTNT$^+$ grafted cardiomyocytes randomly selected from 5 to 10 areas. The numbers were averaged for each transplanted heart. Graft size was measured by calculating the ratio of the GFP$^+$ graft area divided by the entire LV area. Imaging was done using either a Zeiss LSM700 confocal microscope or an EVOS Microscope (ThermoFisher) and analyzed with Zen Blue software (Carl Zeiss) and ImageJ (NIH).

**Quantification and statistical analysis**. All data are represented as mean ± standard error of mean (SEM). Indicated sample sizes (*n*) represent biological replicates including independent cell culture replicates and individual tissue samples. For single cell data, samples size represents the number of cells analyzed from at least three independent experiments. No statistical method was used to predetermine the samples size. Statistical significance was determined using a Student's *t* test (unpaired, two-tailed) or one-way ANOVA with Tukey's multiple comparisons in GraphPad Prism 6 software (GraphPad Software). All statistical parameters are reported in the respective figures and figure legends.

**Reporting summary**. Further information on research design is available in the Nature Research Reporting Summary linked to this article.

## Data availability
The data that support the findings in this study are available within the article and its Supplementary Information files, and from the corresponding author upon request. Raw scRNAseq data generated in this study (day 20 ventricular cardiomyocytes and day 32 immature and mature cardiomyocytes) have been deposited at the GEO database under accession code: GSE152589.

For scRNAseq analysis, the following data sets were used; the Hallmark gene set (h.all. v7.1.symbols) and the TFT_Legacy subset of TFT (c3.tft.tft_legacy.v7.1.symbols); http://www.gsea-msigdb.org/gsea/msigdb/index.jsp

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

## Acknowledgements

We would like to thank past and present members of the Keller lab for their advice and comments on the manuscript and experiments. This research was funded in part by the BlueRock Therapeutics. SF received post-doctoral funding support from Ted Rogers Center for Heart Research. We would like to thank members of the UHN/SickKids flow cytometry sorting facility, the UHN Pathology Research Program, the Princess Margaret Genomics Center at UHN, and the Advanced Optical Microscopy Facility at the Princess Margaret Cancer Research Tower all in Toronto, Ontario, Canada for their technical expertise. We would like to thank R. Hamilton (Sick Kids, Toronto, ON, Canada) for assistance in obtaining fetal tissue samples. We would also like to thank Henry Hong and Audrey Chong of the Imaging Facility in the Department of Cell and Systems Biology at the University of Toronto for their technical expertise in the sample preparation and the use of the electron microscope.

## Author contributions

S.F., I.F. conceived the project, performed experiments, analyzed data, and wrote the manuscript. O.M. and S.S.N. performed the measurement of contraction force. D.W., T.T., D.Y., B.B., J.L., S.P., and G.B. designed and analyzed scRNA-seq data. A.M. performed the transplantation. W.D. and M.A.L. performed the measurement of $Ca^{2+}$ transient and advised on the transplantation experiment. G.M.K. designed the project and wrote the manuscript.

## Competing interests

G.M.K. is a scientific co-founder and paid consultant for BlueRock Therapeutics LP, a paid consultant for VistaGen Therapeutics and a board member of Anagenesis Biotechnologies. M.A.L. is founding investigator and paid consultant for BlueRock Therapeutics. S.P. is a paid consultant for BlueRock Therapeutics. All other authors declare no competing interests.
