## [Peer Review File · Nature Communications]

REVIEWER COMMENTS

Reviewer #1 (Remarks to the Author):

Funakoshi et al describes the novel protocol generating mature compact ventricular cardiomyocytes from hPSCs, which displays activation of fatty acid oxidation, high mitochondrial mass, well-defined sarcomere structures and enhanced contraction force. By usual differentiation protocols, hPSC-derived cardiomyocytes are known to have immature phenotype, which limits the usage for disease modeling and drug testing. However, the maturation strategy proposed in this study clearly showed mature phenotypes of not only ventricular cardiomyocytes but also atrial cardiomyocytes, therefore this protocol would enhance the availability for the usage of the hPSC-derived cardiomyocytes. There are a few comments itemized below.

1. The authors performed all experiments with only one ES cell line. It is possible that any one line can have unique features that may impact the results. The results would be even more convincing if comparable findings were demonstrated with an independent line(s).
2. The authors transplanted freshly isolated matured population of CMs and cryopreserved immature population of CMs and compared graft size, proliferation, Cx43 expression, and maturation at 2 weeks post transplantation. This is rather elusive. Graft size should have been larger if they transplanted freshly isolated immature CMs. Furthermore, histological analysis was performed only 2 weeks after cell transplantation, which does not allow definitive conclusion about the graft characteristics especially when proliferative capacity was different between the two populations.
3. Another issue with regards to CM proliferation assays is that using Ki67 alone would not fully report on CM proliferation and cell division. As these cells could merely synthesize DNA without actual undergoing cytokinesis and division. Further late cytokinesis marker staining would be necessary to accurately address proliferation of these cells.

Reviewer #2 (Remarks to the Author):

In this study, the authors conduct an in-depth assessment of the differentiation of induced pluripotent stem cells (iPS) to mature cardiomyocytes. Extensive single cell (sc) RNA sequencing of the compact population of ventricular myocytes during development led to the delineation of certain cell markers for this subset of ventricular myocytes. Thereafter, human iPS culture systems were largely used to define a cocktail of agents that would first promote proliferation and, thereafter, differentiation to cardiac myocytes using the gene signatures defined in vivo along with markers of metabolic and contractile maturation. The optimal cocktail consisted of T3, dexamethasone, a PPARalpha agonist, and fatty acids. Notably, all of these factors have been shown alone, or in some combination by other groups, to promote cardiac myocyte maturation. Additional scRNA-seq studies of the hiPS-CM population after exposure to a proliferative cocktail (Wnt agonist) followed by the maturation cocktail revealed a number of markers of metabolic maturation including CD36 which could be used as a surface marker for flow sorting. Exposure of the mature cells to pathologic hypertrophy agonist (isoproterenol) downregulated the maturation markers. Lastly, there was some attempt to assess the in vivo engraftment potential of the "mature" cardiac myocytes in nude mice. In a short term analysis, the mature cardiomyocytes engrafted the in vivo heart and exhibited evidence of connexin 43 expression to a greater extent than immature cardiomyocytes.

This is an exhaustive and careful analysis of the differentiation of hPSC to "mature" cardiac myocytes that has identified robust markers of the compact ventricular myocyte population and of metabolic

maturation. Whereas most of the concepts and markers are not particularly surprising based on existing literature, there are valuable findings here. However, as outlined below, the work falls rather short in terms of mechanistic underpinnings, as well as issues with the final set of experiments aimed at in vivo early stage translation.

Specific Critique:

1. Given the importance of the fatty acid oxidation endpoint, a more robust measurement of palmitate oxidation rates would be useful. Standard ¹⁴C-palmitate assays can be conducted on cardiomyocytes for this purpose. Consideration could also be given to assessing intermediates of the β -oxidation pathway (acylcarnitines) using targeted metabolomics in the presence of carnitine.

2. Whereas this work presents important information regarding markers and optimized maturation protocol, mechanistic components are largely lacking. An interesting finding in the genomic interrogation relates to the identification of a nuclear receptor, ERRalpha, as a likely player. It would be important to place this in the context of other published data regarding the potential mechanism whereby ERR is involved. Recently, ERRalpha and ERRgamma were identified as key cardiac maturation factors (Circ Res. 2020 Jun 5;126(12):1685-1702). It was surprising this work was not referenced or put in context. In addition, is PPARalpha upstream or downstream or cooperative with ERR? Previous studies have shown that ERR is capable of activating PPARalpha expression (Mol Cell Biol. 2004;24:9079-9091). From a mechanistic standpoint, it would be interesting to determine whether KD of PPARalpha affects maturation in the PSC-CM system.

3. Similar to the point made in #2, are the thyroid receptor or glucocorticoid receptors necessary for cardiac myocyte maturation?

4. The final series of experiments in which the mature cardiomyocytes are engrafted into nude mice seem incomplete. It is known that engraftment of stem cell-derived cardiac myocytes may cause cardiac rhythm disturbances. In addition, there is the potential issue of gradual loss of such cells. The engraftment experiments would be more meaningful with a longer time course that included electrophysiological and, possibly, functional endpoints.

5. This study and several others have implicated the shift to fatty acid oxidation as a driver of maturation. It would be of interest to determine whether the maturation is halted by inhibition of fatty acid oxidation using an agent such as etomoxir.

6. Whereas a number of metabolic markers of maturation were identified by the RNAseq data, there is little information about contractile markers. Previously, for example, the contractile gene isoform TNNI3 has been shown to signify human adult cardiac myocytes. Did this also serve as a marker in this study? Are there other contractile proteins that are robust differentiation signatures similar to the metabolic markers involved in lipid metabolism?

Reviewer #3 (Remarks to the Author):

This study by Funakoshi et al provides an extensive study covering molecular, cell, and organ level analysis of stem cell differentiation into the cardiac lineage focusing on regulation of cell maturation of compact myocardial cells. They provide demonstration of single cell analysis of these cell types together with functional studies in vitro and in vivo to support the study. Overall, the study fits well within a number of previously published studies covering the area of cardiac maturation that reduce the overall novelty of the study.

I think overall, the data are well laid out and have no issues with the quality of the work performed. Indeed, this work is coming from one of the leading labs worldwide who is developing excellent strategies for differentiating cardiac cell types from pluripotent stem cells. In regard to this study, it is unfortunately an area that has recently been well developed by several other labs in the field such as James Hudson and Chuck Murry among others. I find this reassuring as the growing evidence for mechanisms of metabolic cardiac maturation are growing, including work outlined here. While more detailed points are provided below, I think the study and data could be more truncated to streamline data presentation and writing to focus the study on major novel findings of the study as I felt the content were not well tied together and largely reinforced existing knowledge about metabolic mechanisms of cardiomyocyte maturation. The following points outline a number of specific concerns for consideration:

Major concerns

- 1) In figure 1, it is not clear that the authors are controlling a fate specification choice between compact and trabecular myocardium as opposed to toggling the maturation state of cardiomyocytes. For example CHIR, NRG1 and XAV have all been well defined as determinants of cardiomyocyte proliferation and maturation. Their use in this study is consistent with prior work on controlling cardiac maturation/proliferation, but doesn't contribute to the argument of controlling cell fate choices between compact and trabecular myocardium. Furthermore, genes selected to represent compact vs trabecular myocardium are also markers of normal cardiomyocyte maturation. Making the distinction between fate choice and maturation would require more than is provided from the existing data. This data also does not tie well into subsequent data where the trabecular vs compact myocardial fate switch is not a priority point of analysis in the remaining figures – see comments below.
- 2) Do any results from Fig 1 substantially alter the differentiation approach used in Figure 2-7? There is no clear link between how work on Wnt/NRG1 in Fig 1 influence approaches used for differentiation in data presented in Fig 2 making the narrative quite unclear directly from the outset of the results section. More work is required to help the reader understand the novelty of work from Fig 1 and whether if any of it contributes to the progression into the metabolic maturation studies in Fig 2 etc.
- 3) Much of the maturation condition perturbation factors and phenotyping at the molecular and organoid/physiological level as described in Figures 2-4 and supplemental figures has already been shown in respect to cardiac maturation using these or similar combinations of metabolic maturation factors. The authors have cited this relevant work as recently published for example from James Hudson laboratory and recent work from Chuck Murry's lab among others.
- 4) The authors use various computational tools to study the single cell analysis of the mature vs immature cells (Figure 5). They identify ESSRA motif as a potential driver of maturation, numerous clusters identified with states demarcated by their stress or proliferative state – however this is not well justified or functionally developed making the outcomes largely descriptive. The data confirm the findings in previous figures at a genetic level and the results shown could have probably been derived by bulk RNA-seq. Although reasonably represented, the analysis of cell heterogeneity in the single cell data is not sufficiently developed to add substantively to the paper.
- 5) In the cell transplantation assays, the authors state that they used fresh mature cells and cryo-preserved immature cells. Cryopreservation and thawing is a major stress on cells in this assay and is the reason why groups prefer to use fresh cells for injection. This is an issue that could explain the differences in the outcomes. Aside from that, the conclusions shown are perfectly consistent with a mature/immature distinction between the input cell types in previous figures. However, the functional benefit of injecting mature vs immature cells is not shown (but unlikely to be different) and so provides only incremental value to the novelty of the outcomes over what was presented in previous figures.

Minor concerns

- 1) A schematic of the differentiation protocol would be helpful in Fig 1 panel A to orient the reader to the method used without having to refer to historic studies. In general, simplified schematics of the differentiation conditions used would be very helpful in the main figures rather than refer to prior studies or place this info in the supplement.
- 2) There is no QC data on the single cell RNA-seq in terms of mapped reads and cell quantification. This should be provided as a supplement
- 3) MLY2 instead of MYL2 on line 134
- 4) The fibroblast contribution to the tissue analysis is very important as this can have a major impact on functional studies – data as shown in Supp Fig 4F needs to be quantified over replicates to be shown as statistically not significant between groups.
- 5) The single cell analysis of the mature/immature cardiac differentiation data in Figure 5 are confusing. The text indicates that these data are integrated as represented in Figure 5a. The supplemental figure 5b is critical to help orient the reader to the identity of the input samples since you have merged them in the UMAP in Figure 5a. The text is also not helpful because it suggests that marker genes were used to identify the mature cardiomyocyte population (see lines 377-380). I assume the samples were pre-labelled prior to integration so this is computationally defined rather than using marker genes. Clarify the figures and text to make the cell identity clearer for the reader. The numbering of the clusters is only seen in the supp fig 5b making the text difficult to navigate easily – e.g. should use the same nomenclature to demarcate the data in the main figure as used in the text. Can you also provide a larger panel of maturation genes that are familiar to people – esp the isoform changes that are known to underlie maturation like MYH6/7, MYL7/2, TNNT1/3 etc to indicate that some of these fundamental features of maturation are occurring in coordination with other genetic changes represented in the data.
- 6) RT-PCR data in figure 7a is normalised to TBP. What is this gene? Can you confirm there is no intrinsic difference in abundance of this gene between mature and immature cells based on the scRNA-seq data?

Point-by-point responses to the reviewers' comments.

Reviewer #1 (Remarks to the Author):

1. The authors performed all experiments with only one ES cell line. It is possible that any one line can have unique features that may impact the results. The results would be even more convincing if comparable findings were demonstrated with an independent line(s).

Response: We agree that testing a second hPSC line is important and therefore applied the maturation protocol to cardiomyocytes generated from ESI-17 human embryonic stem cells. Our flow cytometric analyses showed that the ESI-17-derived cardiomyocytes responded as those from the HES2 line and upregulated expression of CD36 and LDLR, confirming the utility of this quantitative approach for monitoring cardiomyocyte maturation. Similarly, these cells upregulated the expression of genes associated with different parameters of maturation including sarcomere formation, Ca²⁺ handling, FAO, and mitochondria and ion channel function. Finally, we found that these cells also upregulated the genes that we've identified as the maturation signature, confirming that their expression tracks well with cardiomyocyte maturation. Collectively, these findings shown in Supplementary Figure 7 confirm that our protocol does promote the maturation of cardiomyocytes generated from different hPSCs.

2. The authors transplanted freshly isolated matured population of CMs and cryopreserved immature population of CMs and compared graft size, proliferation, Cx43 expression, and maturation at 2 weeks post transplantation. This is rather elusive. Graft size should have been larger if they transplanted freshly isolated immature CMs. Furthermore, histological analysis was performed only 2 weeks after cell transplantation, which does not allow definitive conclusion about the graft characteristics especially when proliferative capacity was different between the two populations.

Response: We agree with the reviewer's comments and therefore carried out additional in vivo experiments that included the analyses of the engraftment potential of fresh immature cardiomyocytes at 2 weeks post-transplant and the engraftment potential of fresh immature and mature cardiomyocytes 8 weeks following transplantation. These studies provided the following new information that we have included in the revised manuscript. First, we found that the 2-week-old grafts generated by the freshly isolated immature cardiomyocytes did not differ from those derived from the cryopreserved immature cells with respect to size, sarcomere length of the cells, CX43 expression and the presence of proliferating cells. In the revised manuscript, we now show the differences between the grafts from the freshly isolated immature and mature cells at 2 weeks posttransplant. Second, the mature grafts did not change in size between 2 and 8 weeks. The grafts from the immature cells appear to have increased somewhat in size during this timeframe and at 8 weeks were significantly larger than those from the mature cells. This is likely due to the difference in the proportion of proliferating cells in the grafts from the two populations. Third, the cells from both grafts showed evidence of additional maturation between 2 and 8 weeks. This included an increase in sarcomere length in the cells and an increase in the

proportion of CX43 expressing cells. Additionally, the immature grafts also showed a reduction in the number of proliferating cells. Fourth, despite these changes, differences were still detected at the 8 week timepoint between grafts generated from the two populations. The grafts from the mature cardiomyocytes contained cells with longer sarcomeres, had fewer proliferating cells and more CX43 expressing cells than those from the immature cells. Together, these findings indicate that the maturation status of the graft is influenced by the maturation status of the cells used for transplantation. These results are shown in Figure 9 and Supplementary Figure 10.

3. Another issue with regards to CM proliferation assays is that using Ki67 alone would not fully report on CM proliferation and cell division. As these cells could merely synthesize DNA without actually undergoing cytokinesis and division. Further late cytokinesis marker staining would be necessary to accurately address proliferation of these cells.

Response: We agree with this point and have included analyses of phosphorylated histone H3 as a measure of cell proliferation in the grafts. These data are shown in Supplementary Figure 10b, c.

Reviewer #2 (Remarks to the Author):

1. Given the importance of the fatty acid oxidation endpoint, a more robust measurement of palmitate oxidation rates would be useful. Standard ¹⁴C-palmitate assays can be conducted on cardiomyocytes for this purpose. Consideration could also be given to assessing intermediates of the β-oxidation pathway (acylcarnitines) using targeted metabolomics in the presence of carnitine.

Response: As indicated by the reviewer, fatty acid oxidation (FAO) is an important endpoint in our study as it is a distinguishing feature of mature cardiomyocytes. To document the shift to FAO we have used the Seahorse FAO assay to show that our mature cells are dependent on FAO and able to utilize exogenous palmitate efficiently (Figure 2 and 3). We believe these detailed analyses clearly demonstrate that the mature hPSC-derived cardiomyocytes have undergone a characteristic metabolic shift, similar to that observed in maturing cardiomyocytes in the neonate. The experiments suggested by the reviewer are detailed analyses that would provide information on efficiency and robustness of the FAO response, but would not substantially change the outcome of the study. Given this, we feel they are outside the scope of the current study.

2. Whereas this work presents important information regarding markers and optimized maturation protocol, mechanistic components are largely lacking. An interesting finding in the genomic interrogation relates to the identification of a nuclear receptor, ERRα, as a likely player. It would be important to place this in the context of other published data regarding the potential mechanism whereby ERR is involved. Recently, ERRα and ERRγ were identified as key cardiac maturation factors (Circ Res. 2020 Jun 5;126(12):1685-1702). It was

surprising this work was not referenced or put in context. In addition, is PPARalpha upstream or downstream or cooperative with ERR? Previous studies have shown that ERR is capable of activating PPARalpha expression (Mol Cell Biol. 2004;24:9079-9091). From a mechanistic standpoint, it would be interesting to determine whether KD of PPARalpha affects maturation in the PSC-CM system.

Response: We thank the reviewer for bringing this important study to our attention. We have discussed it and referenced it in our revised manuscript. We agree that a better understanding of the role of ERRalpha and its relationship to PPARa signaling would be of interest and could add new insights into the regulation of human cardiomyocyte maturation. To investigate the role of ERRalpha in maturation of the hPSC-derived cardiomyocytes, we first tested the separate components of our maturation cocktail including Palmitate in low glucose media (PAL), Dex+T3 in high glucose media (DT) and PPARalpha agonist in high glucose media (PPARa) on the cells to determine which regulator induced its expression. The findings from this analysis showed that the hormones, Dex+T3 were the primary drivers of ESRRA expression. PPARa agonist alone did not induce expression above the levels observed in the control cells suggesting that it is not functioning upstream of ESRRA in these cardiomyocytes (Figure 6e). For the second set of experiments, we performed siRNA mediated knockdown (KD) of ESRRA to determine if its upregulation is essential for maturation of hPSC-derived cardiomyocytes. These studies showed that inhibition of ESRRA expression resulted in lower levels of expression of a subset of FAO (*FABP3*, *ACSL1*) and sarcomere (*MYL2*, *TCAP*)-related genes compared to the control cells, indicating that ESRRA is involved in aspects of metabolic and structural maturation (Fig. 6f, g). Beyond this, we also observed an increase in Nile red staining and a downregulation of expression of the hormone sensitive lipase *HSL* in the KD cells (Fig. 6h, i). These findings suggest an increase in lipid storage in these cells indicating that ESRRA also plays a role in the utilization of endogenous lipids. While these differences were detected, the levels of many other genes including *PPARa* were not impacted by the downregulation of ESRRA suggesting that this pathway is only part of regulatory machinery that controls cardiomyocyte maturation. The observation that the levels of *PPARa* expression were not impacted suggests that its expression is not directly regulated by ESRRA. The findings from these KD studies are shown in Figure 6f-i, Supplementary Figure 6d, and Supplementary Table 4 of the revised manuscript. Although the reviewer suggested that we perform a KD of PPARa, we felt that the KD of ESRRA would be more informative as we were able to evaluate the role of PPARa signaling in cardiomyocyte maturation through the addition of the agonist GW7647.

3. Similar to the point made in #2, are the thyroid receptor or glucocorticoid receptors necessary for cardiac myocyte maturation?

Response: This is a good point that we can address with the following observations. First, we have confirmed that both the thyroid and glucocorticoid receptors are expressed in our day 18 cardiomyocytes through RT-qPCR analysis. We have included these data in Supplementary Figure 2c. Second, our flow cytometric analyses showed that the combination of Dex and T3 increased the proportion of CD36⁺ cells above that observed in the population treated with Pal/PPARa, indicating that hormonal stimulation does play a role in maturation of hPSC-derived

cardiomyocytes (Figure 2a, Supplementary Figure 2e). Third, the combination of Dex and T3 (DT) does promote some aspects of maturation including the upregulation of some, but not all FAO related genes (Figure 2c), an increase in OCR (Figure 2d, e), structural improvements (Figure 4a-g), and improvements in Ca^{2+} handling (Figure 4i). Taken together, these findings indicate that hormonal stimulation is important for cardiomyocyte maturation and therefore we included Dex and T3 in our maturation cocktail.

4. The final series of experiments in which the mature cardiomyocytes are engrafted into nude mice seem incomplete. It is known that engraftment of stem cell-derived cardiac myocytes may cause cardiac rhythm disturbances. In addition, there is the potential issue of gradual loss of such cells. The engraftment experiments would be more meaningful with a longer time course that included electrophysiological and, possibly, functional endpoints.

Response: The goal of the transplantation experiments using the nude rat model was to determine if the maturation status of the cells used for transplantation would influence the physical/morphological aspects of the grafts. With this goal in mind, we focused on histomorphological analyses that included evaluation of graft size and maturation parameters of the cells within the grafts such as sarcomere length, cell cycle status and CX43 expression. Given the rapid heart rate of the rat (>300/min), hPSC-derived cardiomyocytes are not able to efficiently electrically couple with the host rat myocardium and therefore meaningful electrophysiological and functional analyses are not possible in this model. These analyses are planned in future studies using the guinea pig that does enable electrical integration of hPSC-derived cardiomyocytes¹. We do agree with the reviewer that a longer time course would be very informative and therefore repeated the transplantation experiments and analyzed the animals at 8 weeks post transplantation. These studies provided the following new information that we have included in the revised manuscript. First, neither the mature nor the immature grafts decreased in size between 2 weeks and 8 weeks indicating there was no significant loss of cells during this time. Second, the grafts from both populations showed evidence of maturation over this timeframe. This included an increase in sarcomere length in the cardiomyocytes and an increase in the proportion of CX43 expressing cells in grafts from both cell populations. Additionally, the immature grafts also showed a reduction in the number of proliferating cells. Third, despite these changes, differences were still detected at the 8 week timepoint between grafts generated from the two populations. The grafts from the mature population contained cells with longer sarcomeres, had fewer proliferating cells and more CX43 expressing cells than those from the immature cells. Together, these findings show that the maturation status of the graft is influenced by the maturation status of the cells used for transplantation. These results are shown in Figure 9 and Supplementary Figure 10.

5. This study and several others have implicated the shift to fatty acid oxidation as a driver of maturation. It would be of interest to determine whether the maturation is halted by inhibition of fatty acid oxidation using an agent such as etomoxir.

Response: We agree that further evidence demonstrating that FAO is a driver of cardiomyocyte maturation would be important. To address this, we followed the reviewer's suggestion and

asked if addition of etomoxir (ETO) to the cardiomyocyte populations would impact maturation. For this experiment, we added etomoxir throughout the maturation stage of the protocol (from day 18 to day 32) and then analyzed the end stage population by RT-qPCR. As shown in Figure 4k, we observed a significant reduction in the levels of expression of the mitochondrial genes *CKMT2*, *COX7A1*, and *COX6A2*, the sarcomere genes *MYL2* and *MYOZ2* and the Ca²⁺ handling related gene *ATP2A2* in the cardiomyocytes treated with ETO indicating that FAO does play a role in human cardiomyocyte maturation. However, not all of the changes associated with maturation were inhibited in the ETO-treated cells. Other genes involved in these processes including the sarcomere (*TCAP*, *DES*, *MYOM3*), mitochondria (*ATP5A1*, *ATPIA3*), Ca²⁺ and ion channel (*KCNJ2*, *HRC*, *CALM1*)-related genes, were not affected by the addition of ETO (data not shown), indicating that pathways functioning independent of FAO are also impacting maturation.

6. Whereas a number of metabolic markers of maturation were identified by the RNAseq data, there is little information about contractile markers. Previously, for example, the contractile gene isoform *TNNI3* has been shown to signify human adult cardiac myocytes. Did this also serve as a marker in this study? Are there other contractile proteins that are robust differentiation signatures similar to the metabolic markers involved in lipid metabolism?

Response: The GO analyses presented in Figure 5b (Figure 5e in the revised version) of our original manuscript showed that genes associated with sarcomere organization and muscle contraction were upregulated in mature cells compared to the immature population. We agree with the reviewer that the identification of specific contractile genes that could be included as part of a differentiation signature would be very informative. To address this, we have included a violin plot showing differential expression of a number of different contractile genes including *TNNI1/3*, *MYH6/7*, *MYL2/7*, *MYPN*, *TTN*, *OBSCN*, *ACTN2*, *MYLK3*, *SYNPO2L*, *TCAP*, *MYOZ2*, and *DES*. These data are presented in Supplementary Figure 5b in the revised manuscript. With respect to the switch in *TNNI1/3* isoform expression, we found that the expression of *TNNI1* was downregulated in the mature cells compared to the immature cells. However the levels of *TNNI3* were comparable between the two populations. We did observe upregulation of *TNNI3* expression in the 8-week-old but not in the 2-week-old grafts from both the mature and immature cells (Supplementary Figure 10f). These findings indicated that this aspect of maturation occurs beyond the time-frame of our in vitro cultures and/or requires stimuli not provided within the EBs. Further experiments will be required to identify the specific regulation of *TNNI3*.

Reviewer #3 (Remarks to the Author):

Major concerns

1) In figure 1, it is not clear that the authors are controlling a fate specification choice between compact and trabecular myocardium as opposed to toggling the maturation state of

cardiomyocytes. For example CHIR, NRG1 and XAV have all been well defined as determinants of cardiomyocyte proliferation and maturation. Their use in this study is consistent with prior work on controlling cardiac maturation/proliferation, but doesn't contribute to the argument of controlling cell fate choices between compact and trabecular myocardium. Furthermore, genes selected to represent compact vs trabecular myocardium are also markers of normal cardiomyocyte maturation. Making the distinction between fate choice and maturation would require more than is provided from the existing data. This data also does not tie well into subsequent data where the trabecular vs compact myocardial fate switch is not a priority point of analysis in the remaining figures – see comments below.

Response: We have provide clarification at the beginning of the Results section indicating our initial goal was to generate cardiomyocytes that display characteristics of compact ventricular cells to be able to model disease that affect these cells and to develop new cell based therapies to treat them. We also add a sentence at the beginning of the second section of the results stating that we used the compact ventricular cardiomyocyte as the target population to develop and optimize the protocol that promotes maturation. With respect to the issue regarding the generation of compact and trabecular cells, studies in model organisms have shown that these cardiomyocyte subtypes can be distinguished by the differential expression of the following genes: HEY2, MYCN, HIF1A and CCND2 in compact and NPPA, BMP10, SCN5A and IRX3 in trabecular. The reviewer is correct in pointing out that some, but not all of these genes are expressed in different patterns as the heart matures. However, in the context of the early developing heart, their expression tracks faithfully with the two different cardiomyocyte cell types. We believe that HEY2 and MYCN are compact specific as we have not found any literature showing that they are expressed in the trabecular population at any stage of development. Our initial single cell RNA-seq analyses showed that the ventricular cardiomyocyte population generated with our previously published protocol consisted predominantly of HEY2⁺ compact cardiomyocytes along with a small subset of ANF⁺ trabecular cardiomyocytes suggesting that those conditions promote the development of both populations. To be able to generate compact cardiomyocyte populations devoid of trabecular cells, we again turned to studies in model organisms that showed that these two cardiomyocyte subtypes are specified through different signaling pathways. Neuregulin (NRG1) secreted by the endocardium is the primary driver of trabecular fate whereas Wnt and IGF2 secreted by the epicardium have been shown to play important role in the generation of compact cardiomyocytes. Notably, our scRNA-seq analyses also identified Wnt signaling as a potential regulator of the compact fate. We show that treatment of day 10 cardiomyocytes with CHIR/IGF2 promoted the development of a population that expressed compact markers with significantly reduced levels of the trabecular markers and that the addition of NRG1 induced a cardiomyocyte population that displayed the opposite expression patterns. These findings provide strong evidence that the regulation of human trabecular and compact development *in vitro* recapitulate that observed *in vivo* and strongly support the interpretation that we have generated the two subtypes of cardiomyocytes from hPSCs (Figure 1h-j). In our revised manuscript, we have added the expression patterns of several additional differentially expressed genes between the CHIR/IGF2-treated compact and the NRG1-treated trabecular populations (Supplementary Figure 1e). The ability to direct a compact vs a trabecular fate is a new and important finding, as these cells represent distinct cardiomyocyte subtypes that carry out different functions. We agree with the reviewer that several of these pathways promote cardiomyocyte proliferation and have

referenced these studies in our manuscript. We do show that the combination of Wnt (CHIR) and IGF2 signaling promotes proliferation of the compact populations (Figure 1f, g), recapitulating the role of these pathways in the developing heart in vivo.

2) Do any results from Fig 1 substantially alter the differentiation approach used in Figure 2-7? There is no clear link between how work on Wnt/NGR1 in Fig 1 influence approaches used for differentiation in data presented in Fig 2 making the narrative quite unclear directly from the outset of the results section. More work is required to help the reader understand the novelty of work from Fig 1 and whether if any of it contributes to the progression into the metabolic maturation studies in Fig 2 etc.

Response: As stated in our response above (#1), the goal of the experiments outlined in Figure 1 was to develop a strategy to generate compact cardiomyocytes for the maturation studies. They were not intended to influence or inform the approaches used for maturation, but rather to provide an appropriate target population. The novelty of the work in Figure 1 is the demonstration that it is possible to specify a compact or trabecular cardiomyocyte fate from immature hPSC-derived cells through the manipulation of signaling pathways that have been shown to regulate these fates in vivo.

3) Much of the maturation condition perturbation factors and phenotyping at the molecular and organoid/physiological level as described in Figures 2-4 and supplemental figures has already been shown in respect to cardiac maturation using these or similar combinations of metabolic maturation factors. The authors have cited this relevant work as recently published for example from James Hudson laboratory and recent work from Chuck Murry's lab among others.

Response: We agree with the reviewer that others have shown that some of the factors we have used are able to promote some degree of metabolic maturation in hPSC-derived cardiomyocytes. We have referenced many of these studies in our manuscript. Our studies goes beyond what other have shown and provide the following new information: i) strategies for the specification of compact vs trabecular cardiomyocytes, ii) a comparison of the effectiveness of different combinations of factors on maturation using quantitative flow cytometry analyses, this approach showed that the combination of PPAR α signaling with hormonal and FA stimulation was optimal and promoted efficient metabolic maturation of the majority of the compact cardiomyocytes; we have included a new table in the revised manuscript that summarizes the effect of the different stimuli on cardiomyocyte maturation (Table 1), iii) demonstration that transient treatment with the maturation factors is required for the generation of cells that can oxidize exogenously provided FAs, iv) global molecular profiling showing efficiency of maturation, the identification of LDLR expression as a maturation marker and the identification of a molecular signature of metabolically mature cardiomyocytes, v) demonstration that the maturation protocol promotes maturation of atrial cardiomyocyte, vi) demonstration that the mature ventricular cells can be used to model key aspects of heart failure in vitro and vii) demonstration that the mature cells generate grafts that contain more mature cells than those from immature cells.

4) The authors use various computational tools to study the single cell analysis of the mature vs immature cells (Figure 5). They identify ESSRA motif as a potential driver of maturation, numerous clusters identified with states demarcated by their stress or proliferative state – however this is not well justified or functionally developed making the outcomes largely descriptive. The data confirm the findings in previous figures at a genetic level and the results shown could have probably been derived by bulk RNA-seq. Although reasonably represented, the analysis of cell heterogeneity in the single cell data is not sufficiently developed to add substantively to the paper.

Response: Our single cell RNA-seq analyses has provided the following new information that we feel is an important contribution to the paper: i) demonstrated that the majority of the cells in the mature population express markers of metabolic maturation (*CD36*, *FABP3*, *ACSL1*, *CKMT2* and *COX6A2*; Figure 5c) indicating that protocol is efficient, ii) identified LDLR as a marker of metabolic maturation providing a second surface marker to monitor maturation by flow cytometry, iii) identified a molecular signature of metabolically mature cells that can be used to monitor maturation and compare populations generated by different groups, iv) identified ESSRA as a potential driver of maturation. To further assess the function of ESSRA we performed siRNA mediated knockdown (KD) to determine if its upregulation is essential for maturation of hPSC-derived cardiomyocytes. These studies showed that inhibition of ESSRA expression resulted in lower levels of expression of a subset of FAO (*FABP3*, *ACSL1*) and sarcomere (*MYL2*, *TCAP*)-related genes compared to the control cells, indicating that ESSRA is involved in aspects of metabolic and structural maturation. Beyond this, we also observed an increase in Nile red staining and a downregulation of expression of the hormone sensitive lipase *HSL* in the KD cells. These findings suggest an increase in lipid storage in these cells indicating that ESSRA also plays a role in the utilization of endogenous lipids (Figure 6f-i, Supplementary Figure 6d).

5) In the cell transplantation assays, the authors state that they used fresh mature cells and cryopreserved immature cells. Cryopreservation and thawing is a major stress on cells in this assay and is the reason why groups prefer to use fresh cells for injection. This is an issue that could explain the differences in the outcomes. Aside from that, the conclusions shown are perfectly consistent with a mature/immature distinction between the input cell types in previous figures. However, the functional benefit of injecting mature vs immature cells is not shown (but unlikely to be different) and so provides only incremental value to the novelty of the outcomes over what was presented in previous figures.

Response: The goal of the transplantation experiments using the nude rat model was to determine if the maturation status of the cells used for transplantation would influence the physical/morphological aspects of the grafts. Given the rapid heart rate of the rat (>300/min), hPSC-derived cardiomyocytes are not able to efficiently electrically couple with the host rat myocardium and therefore meaningful electrophysiological and functional analyses are not possible in this model. We agree with the concerns of comparing fresh and cryopreserved cells and therefore carried out additional experiments to compare the engraftment potential of fresh immature and mature cells at 2 weeks post-transplant. Additionally, we set up an independent

experiment to compare the engraftment potential of the mature and immature populations at 8 weeks. The findings from these studies provided the following new information that we have included in the revised manuscript. First, we found that the 2-week-old grafts generated by the freshly isolated immature cardiomyocytes did not differ from those derived from the cryopreserved cells with respect to size, sarcomere length of the cells, CX43 expression and the presence of proliferating cells. In the revised manuscript, we now show the differences between the grafts from the freshly isolated immature and mature cells at 2 weeks posttransplant. Second, grafts from both populations showed evidence of maturation over this timeframe. This included an increase in sarcomere length in the cells and an increase in the proportion of CX43 expressing cells in grafts. The immature grafts also showed a reduction in the number of proliferating cells. Third, despite these changes, differences were still detected at the 8 week timepoint between grafts generated from the two populations. The grafts from the mature population contained cells with longer sarcomeres, had fewer proliferating cells and more CX43 expressing cells than those from the immature cells. Together, these findings indicate that the maturation status of the graft, even 8 weeks following transplantation, is influenced by the maturation status of the input cell population. These results are shown in Figure 9 and Supplementary Figure 10.

We believe that the differences observed, in particular those at the early time points, will impact graft behavior and function and predict that the grafts from the mature cells will integrate more efficiently and provide improvements in function more rapidly than those from the immature cells. These functional analyses will be carried out in future studies in the guinea pig model that does enable electrical integration of human cardiomyocytes¹.

Minor concerns

1) A schematic of the differentiation protocol would be helpful in Fig 1 panel A to orient the reader to the method used without having to refer to historic studies. In general, simplified schematics of the differentiation conditions used would be very helpful in the main figures rather than refer to prior studies or place this info in the supplement.

Response: We have added a schematic to Figure 1a. We additionally put the final differentiation protocol in Figure 3e.

2) There is no QC data on the single cell RNA-seq in terms of mapped reads and cell quantification. This should be provided as a supplement

Response: We have included QC data in Supplementary table 1.

3) MLY2 instead of MYL2 on line 134

Response: We made this change in the revised manuscript.

4) The fibroblast contribution to the tissue analysis is very important as this can have a major impact on functional studies – data as shown in Supp Fig 4F needs to be quantified over replicates to be shown as statistically not significant between groups.

Response: We have added the quantification in Supplementary Figure 4g.

5) The single cell analysis of the mature/immature cardiac differentiation data in Figure 5 are confusing. The text indicates that these data are integrated as represented in Figure 5a. The supplemental figure 5b is critical to help orient the reader to the identity of the input samples since you have merged them in the UMAP in Figure 5a. The text is also not helpful because it suggests that marker genes were used to identify the mature cardiomyocyte population (see lines 377-380). I assume the samples were pre-labelled prior to integration so this is computationally defined rather than using marker genes. Clarify the figures and text to make the cell identity clearer for the reader. The numbering of the clusters is only seen in the supp fig 5b making the text difficult to navigate easily – e.g. should use the same nomenclature to demarcate the data in the main figure as used in the text. Can you also provide a larger panel of maturation genes that are familiar to people – esp the isoform changes that are known to underlie maturation like MYH6/7, MYL7/2, TNNI1/3 etc to indicate that some of these fundamental features of maturation are occurring in coordination with other genetic changes represented in the data.

Response: We agree and have modified the figures (Figure 5a-d) and the text in the revised manuscript. Additionally, we added the expression profiles of structural and electrophysiological genes related to the maturation in scRNAseq in Supplementary Figure 5b. We have included a discussion of the observed patterns of expression with TNNI1/3 and MYH6/7 in the main text as followed,

“This analysis showed that many of these genes were upregulated in the mature cells compared to the immature population. Several notable exceptions were observed. The first is in the switch in the TNNI1/3 isoforms known to occur during cardiomyocyte maturation. While our mature cells showed lower levels of the fetal form TNNI1 than the immature cells, the levels of the adult isoform, TNNI3 were comparable in the 2 populations. The reason for this is currently not clear, as the mature cells have upregulate many other sarcomere genes. The second is in the expression patterns of MYH6 and 7 that are opposite of what we expect. One reason for this may be the presence of T3 in our maturation cocktail as it has been shown to induce MYH6 and inhibit MYH7 expression” in the result section, (Page 14, line 1-9) and

“Although broad changes in maturation were observed in the PPDT/PAL-treated cells, they failed to undergo the switch in expression of the TNNI isoforms *in vitro* and only upregulated expression of the adult form following transplantation *in vivo*. One interpretation of these findings is that the upregulation of TNNI3 expression requires mechanical and/or other stimuli that are not be provided in the EBs *in vitro*.” in the discussion. (Page 22, line 19-23).

6) RT-PCR data in figure 7a is normalised to TBP. What is this gene? Can you confirm there is no intrinsic difference in abundance of this gene between mature and immature cells based on the scRNA-seq data?

Response: TBP, TATA-BOX Binding Protein, is the commonly used housekeeping gene. We have confirmed comparable expression levels of TBP between in the immature cells and mature cells as shown in the violin plot below.

Reference

1. Shiba, Y. *et al.* Human ES-cell-derived cardiomyocytes electrically couple and suppress arrhythmias in injured hearts. *Nature* **489**, 322-325 (2012).

REVIEWERS' COMMENTS

Reviewer #1 (Remarks to the Author):

The authors worked hard to respond reviewers' comments, resulting in substantial improvement of the manuscript. I have no further comments.

Reviewer #2 (Remarks to the Author):

None

Reviewer #3 (Remarks to the Author):

The authors have provided a number of new results to the study that have addressed the majority of my concerns. My only ongoing comment, as indicated in my previous review, is that many of the major experiments/findings of this study (metabolism, cell maturation, cell transplantation) have all been demonstrated by various other groups previously which diminishes the novelty of the study. The scope of the work is extensive but not entirely cohesive making the narrative a bit challenging to follow. If the study should progress toward publication, I recommend reducing 9 main figures and draw out only the major new findings that this study contributes to the literature. Most notably, I still consider that the trabecular vs compact myocardial differentiation the most important component of the work with the remainder of the experimental work largely incremental given previous studies in this area.